# Inhaling Peppermint Essential Oil as a Promising Complementary Therapy in the Treatment of Nausea and Vomiting

**DOI:** 10.3390/jcm14145069

**Published:** 2025-07-17

**Authors:** Dorottya Gergő, Gantsetseg Garmaa, Andrea Tóth-Mészáros, Uyen Nguyen Do To, Péter Fehérvári, Andrea Harnos, Péter Hegyi, Rita Nagy, András Bánvölgyi, Attila Ványolós, Dezső Csupor

**Affiliations:** 1Department of Pharmacognosy, Semmelweis University, 1085 Budapest, Hungary; gergo.dorottya@gmail.com (D.G.); vanyolos.attila@semmelweis.hu (A.V.); 2Centre for Translational Medicine, Semmelweis University, 1085 Budapest, Hungary; gantsetseg.garmaa@gmail.com (G.G.); a.toth.mesz@icloud.com (A.T.-M.); touyenpro127@gmail.com (U.N.D.T.); peter.fehervari.tmk@gmail.com (P.F.); harnosandrea@gmail.com (A.H.); hegyi2009@gmail.com (P.H.); nagyrita003@gmail.com (R.N.); banvolgyi.andras@gmail.com (A.B.); 3Department of Pathology, School of Medicine, Mongolian National University of Medical Sciences, Ulan-Bator 14210, Mongolia; 4András Pető Faculty, Semmelweis University, 1125 Budapest, Hungary; 5Department of Biostatistics, University of Veterinary Medicine, 1078 Budapest, Hungary; 6Institute for Translational Medicine, Medical School, University of Pécs, 7624 Pécs, Hungary; 7Institute of Pancreatic Diseases, Semmelweis University, 1083 Budapest, Hungary; 8Heim Pál National Pediatric Institute, 1089 Budapest, Hungary; 9Department of Dermatology, Venereology and Dermato-Oncology, Faculty of Medicine, Semmelweis University, 1085 Budapest, Hungary; 10Center for Pharmacology and Drug Research & Development, Semmelweis University, 1089 Budapest, Hungary; 11Institute of Clinical Pharmacy, University of Szeged, 6725 Szeged, Hungary

**Keywords:** peppermint oil, nausea, vomiting, postoperative, pregnancy, chemotherapy

## Abstract

**Background**: Nausea and vomiting frequently occur during postoperative recovery, chemotherapy, and pregnancy. While peppermint oil is traditionally used to relieve these symptoms, its efficacy remains uncertain. This systematic review and meta-analysis evaluates the efficacy of peppermint oil inhalation for postoperative (PONV), chemotherapy-induced (CINV), and pregnancy-related nausea and vomiting (NVP). **Methods**: Following PRISMA guidelines, we searched five databases (Scopus, Embase, the Cochrane Central Register of Controlled Trials, MEDLINE, and Web of Science) in November 2022, with an update in December 2024. Randomised controlled trials were included, comparing peppermint oil inhalation to a control in patients with PONV, CINV, and NVP. Separate meta-analyses were conducted for each patient group using R, focusing on the severity of the nausea and vomiting. **Results**: Nineteen RCTs were included. In three PONV studies, peppermint oil inhalation was associated with a reduction in nausea 2 to 6 h after the intervention (MD: −0.60 points, 95% confidence interval (CI): −0.77 to −0.44, *p* = 0.004). In three NVP studies, daily peppermint oil treatment was linked to lower symptom severity at 48 h (MD: −0.51, 95% CI: −0.78 to −0.24, *p* = 0.015) and 96 h (MD: −0.68, 95% CI: −1.09 to −0.27, *p* = 0.019). In three CINV studies, peppermint oil inhalation appeared to reduce symptoms at all time points, with the most notable reduction at 48 h (MD: −2.23, 95% CI: −3.13 to −1.34, *p* < 0.001) and 72 h (MD: −2.41, 95% CI: −3.96 to −0.86, *p* = 0.010). **Conclusions**: Peppermint oil inhalation may be a promising complementary therapy for reducing nausea and vomiting in postoperative, chemotherapy, and pregnancy settings.

## 1. Introduction

Nausea and vomiting (NV) are frequent symptoms that can occur in various conditions, including the postoperative period, as a result of chemotherapy treatment, or during pregnancy. These symptoms result from continuous interactions among different parts of the gastrointestinal tract, including the enteric nervous system, the central nervous system, and the autonomic nervous system [1]. Postoperative nausea and vomiting (PONV) are common adverse events of anaesthesia and surgery, occurring in approximately one out of three postoperative patients. The risk is higher in those with established risk factors, including female sex, non-smoking status, a history of PONV or motion sickness, or the use of postoperative opioids. Younger age is also associated with increased risk, with risk decreasing as age advances [2,3,4]. The type of surgery and the type of anaesthesia, particularly the use of volatile anaesthetics, further increase risk, while regional techniques are generally associated with a lower incidence of PONV [5,6,7]. In particular, for patients undergoing vitreoretinal surgeries, especially those with diabetes, PONV can lead to serious complications, such as suprachoroidal haemorrhage, potentially resulting in vision impairment [8,9]. The relationship between obesity and PONV is controversial. Most evidence indicates that obese patients are not at increased risk and may experience a lower incidence, while underweight patients (BMI < 19) may have a higher risk [10,11]. The incidence of PONV may be further reduced by employing Surgical Pleth Index-guided opioid-free anaesthesia [12], or even lowered below 10% with Adequacy of Anaesthesia guidance [13]. Nausea and vomiting during pregnancy (NVP) typically begin in the first trimester and may last throughout the pregnancy. NVP affects around 70–80% of pregnant women. It is thought to be caused by hormonal changes, particularly by an increase in human chorionic gonadotropin (hCG) and oestrogen levels [14,15]. NVP can cause discomfort, dehydration, and malnutrition and affect the quality of life of women [16]. Chemotherapy-induced nausea and vomiting (CINV) is a common and distressing adverse event affecting 40–80% of chemotherapy patients [17]. CINV may compromise patient compliance with the chemotherapy regimen [18]. Consequently, CINV can decrease patients’ quality of life [19].

Current pharmaceutical treatments may be inadequate to treat these symptoms and are often associated with unpleasant adverse events [20,21]. For instance, the 5-HT3 and NK1 receptor antagonists may cause constipation or headaches [22]. In pregnancy, antiemetic use is limited due to safety concerns [23]. Consequently, alternative therapies for NV are needed that may be more effective with fewer side effects. One alternative tool may be aromatherapy, which involves inhaling essential oils (EOs) and is considered a mild therapeutic modality without severe side effects [24].

The inhalation of peppermint essential oil can help alleviate nausea and vomiting through several mechanisms. Peppermint oil has been shown to block serotonin receptors in the gastrointestinal tract that trigger the nausea and vomiting reflex [25]. The active compounds of peppermint oil are menthol (35–45%) and menthone (10–30%), which exhibit antispasmodic effects [26]. These compounds help relax the smooth muscles of the gastrointestinal tract [27].

Several randomised controlled trials (RCTs) have been conducted to evaluate the effect of peppermint on PONV, NVP, and CINV. However, its clinical efficacy has not been established yet [28,29,30,31,32,33]. Several reports indicate that the inhalation of peppermint oil has no serious adverse effects on pregnant women [34]. According to the Australian Therapeutic Goods Administration Classification for Drugs in Pregnancy, peppermint oil is classified as category B2 for use in pregnancy. There is no evidence of an increase in birth defects or other harmful effects on the developing embryo for pharmaceuticals in category B2 [35].

## 2. Materials and Methods

We followed the recommendations of the Preferred Reporting Items for Systematic Reviews and Meta-Analyses (PRISMA) 2020 guideline [36] and the Cochrane Handbook [37]. No ethical approval was required, as all data were published in peer-reviewed journals. The review protocol was registered in the International Prospective Register of Systematic Reviews (PROSPERO) database (CRD42022379103) on 7 December 2022.

### 2.1. Eligibility Criteria

We used the PICO framework to answer our clinical questions. The population (P) included adult patients (>18 years old) with NV symptoms postoperatively, during pregnancy, or as a result of chemotherapy. The patients in the intervention group (I) received peppermint oil by inhalation; the patients in the control group (C) received a placebo. The primary outcome (O) was efficacy, characterised by changes in the severity of NV.

### 2.2. Information Sources

Our systematic search was conducted on 26 November 2022 across five databases (Scopus, Embase, the Cochrane Central Register of Controlled Trials (CENTRAL), MEDLINE (via NCBI PubMed), and Web of Science), with an update on 14 December 2024.

### 2.3. Search Strategy

During the systematic search, the following search key was used: ((peppermint) OR (Mentha piperita)) AND ((nausea) OR (vomiting)), without any filtering options or restrictions. There were no language restrictions. Details can be found in Appendix A.

### 2.4. Selection Process

In our study, RCTs were eligible. Studies were excluded from the systematic review and meta-analysis if (1) they did not meet the inclusion criteria; (2) the intervention combined peppermint oil with other treatments; (3) the study design was a conference abstract, case report, case series, or article with no original data; or (4) desired outcomes could not be obtained from the published findings or upon request from the corresponding author.

The search results were first exported to the EndNote X9 citation manager (Clarivate Analytics, Philadelphia, PA, USA). Duplicates were removed automatically and (DG) manually. Then, two independent authors performed selection by title and abstract. Next, full-text selection was performed using the online screening tool Rayyan (Qatar Computing Research Institute, Hamad Bin Khalifa University, Doha, Qatar; https://www.rayyan.ai) (access on 10 July 2025) [38], according to the inclusion criteria (DG and ATM). In cases of disagreements, a third author (GG) made the final decision. Cohen’s kappa coefficient was calculated at each selection step to evaluate the level of agreement between the authors. The references from eligible studies were screened for eligibility manually and with an automated citation chaser [39].

### 2.5. Data Collection Process

The data were collected from eligible articles by two independent investigators (DG and UNDT). The data were extracted manually from eligible articles, and then the investigators cross-checked each other’s datasets to ensure precision. Disagreements were resolved by consensus. We used Plot Digitizer (https://plotdigitizer.com (accessed by 14 June 2023), Version 2.6.9, 2020) to read data from plots. Microsoft Excel (Microsoft, Office 365, Redmond, WA, USA) was used for data collection. Study authors were approached to request any missing data.

### 2.6. Data Items

The following data were extracted: study characteristics (first author, year of publication, and country), study population (sample size, gender, and age), intervention type and details (herbal medicine type, dose, and duration), and the severity of the NV as the primary outcome.

### 2.7. Risk of Bias Assessment

Two authors (DG and UNDT) independently assessed the risk of bias using the Cochrane risk-of-bias tool (RoB2) Excel tool, version 9 (Cochrane Methods Group, London, UK; https://www.riskofbias.info/) (access on 10 July 2025) [40]. Disagreements were resolved by consensus. The authors evaluated the bias through domains, such as bias due to randomisation, deviations from the intended intervention, missing data, outcome measurement, and the selection of reported results. The risk assessment conclusion categorised the risk of bias as ‘low,’ ‘some concerns’, or ‘high’.

### 2.8. Quality of Evidence

We used the GRADEpro (Guideline Development Tool) [41] tool to evaluate the quality of evidence. Two authors (DG and GG) independently performed the grading of the level of evidence. Each outcome was rated according to the following: risk of bias, inconsistency, indirectness, imprecision, publication bias, presence of a large effect, dose-dependent response, and plausible confounders (‘not serious’, ‘serious’, or ‘very serious’). The final certainty of the evidence was categorised as ‘very low’, ‘low’, ‘moderate’, or ‘high’.

### 2.9. Synthesis Methods and Statistical Analysis

Both qualitative and quantitative syntheses of the data were performed. At least three studies with poolable effect sizes were required for statistical analysis. Meta-analyses were conducted using the ‘meta’ [42] and ‘dmetar’ [43] packages in the R statistical environment (version 4.1.1 (R Foundation for Statistical Computing, Vienna, Austria; https://www.R-project.org/) (access on 10 July 2025) [44].

To account for differences in pathophysiology and patient populations, we conducted separate meta-analyses for PONV, CINV, and NVP. The studies included slightly different numerical scales to quantify the severity of nausea and discomfort. We managed to convert all applied scales to 0–10. Consequently, we could use the mean differences (MDs) between the intervention and control groups’ scales as an effect size measure (with 95% confidence intervals). We applied the same approach to other outcomes in the meta-analysis, such as NV scores and overall satisfaction levels. We extracted the sample size, means, and corresponding standard deviations (SDs) from the studies separately for each group to calculate study and pooled MDs. After conversion to a scale of 0–10, the mean values in the control group were subtracted from the mean values of the experimental group. If quartiles were given instead of the means, SDs, or standard errors of the mean (SEMs), the Luo and Shi methods were used [45,46] to estimate the means and SDs, as implemented in the meta R package (version 4.1.1 (R Foundation for Statistical Computing, Vienna, Austria; https://www.R-project.org/).

The random-effects model was chosen for the meta-analyses. The inverse variance weighting method was used to calculate the pooled MDs. To estimate the heterogeneity variance measure (τ^2^), we used the restricted maximum-likelihood estimator with the Q profile method for confidence intervals [47,48]. In individual studies, the t-distribution-based method was used for the CI of MD calculation.

The subgroup analysis was based on different follow-up intervals, which were determined after data had been extracted, as the data suggested this structure. In the subgroup analysis, we used a fixed-effects “plural” model (aka mixed-effects model). We assumed different τ^2^ values in the subgroups.

Statistical heterogeneity was assessed using the Cochrane Q test and I^2^ values [48]. Small study publication bias was evaluated by visual inspection of funnel plots and by calculating the classical Egger’s test *p*-value [49]. In addition, analyses of outlier and influential points were conducted, following the methodologies proposed by Harrer et al. (2021) [43] and Viechtbauer and Cheung (2010) [50]. The results were considered statistically significant if the pooled CI did not contain the null value.

We summarised the findings from the meta-analysis on forest plots. Due to a low number of studies, we did not report prediction intervals. The time-related pattern in the NV scores by study and treatment was plotted to visualise differences in trends between the intervention and control groups.

## 3. Results

### 3.1. Search and Selection

Our systematic search identified 1547 articles: Scopus (*n* = 970), EMBASE (*n* = 341), Cochrane Library (Trials) (*n* = 93), Web of Science (*n* = 78), and PubMed (*n* = 65). After duplicate removal (*n* = 483), title and abstract selection (*n* = 1064), and full-text selection (*n* = 71), we identified 16 eligible articles (Figure 1). Seventeen publications were identified through an electronic search of databases, and two additional publications were identified by the “citationchaser” tool [39] (eleven postoperative studies [28,33,51,52,53,54,55,56,57,58,59], five chemotherapy studies [30,31,60,61,62], and three pregnancy studies [29,32,63]). After our renewed systematic search, we found three more eligible studies. We included 14 studies in the quantitative analysis (postoperative patients: [33,51,52,53,54,55,56,57], chemotherapy patients: [30,31,60], and pregnant women: [29,32,63]). Five studies were included in the qualitative analysis (postoperative patients: [28,58,59] and chemotherapy patients: [61,62]). The search results and the selection process are summarised in the PRISMA flowchart 2020 (Figure 1).

### 3.2. Baseline Characteristics of Studies Included

The baseline characteristics of the studies included in the meta-analysis are detailed in Table 1. The types and original ranges of the NV measurement tools are detailed in Appendix A. Appendix A include the extended baseline characteristics tables for all three subgroups.

The postoperative studies were conducted across various countries, including Iran, the USA, Turkey, South Korea, and the UK. The sample sizes varied, ranging from 18 patients to over 1000. The mean age of patients ranged from the early 30 s to the mid-70s, with most studies focusing on middle-aged adults. Gender distribution varied, with some studies including only female participants, while others were more balanced. Surgical procedures were diverse, with some studies involving abdominal, open-heart, laparoscopic, orthopaedic, and gynaecological procedures. Most studies used peppermint essential oil as an intervention, with a few using peppermint spirit. Peppermint spirit is a pharmacy-grade, alcohol-based solution containing approximately 82% ethyl alcohol, peppermint oil, peppermint leaf extract, and purified water [57,58]. The follow-up periods ranged from 5–10 min up to 72 h.

The chemotherapy studies were conducted in Turkey, Iran, the USA, and Indonesia. The sample sizes ranged from 80 to 285 patients. The mean age of patients was between 40 and 60 years. Most studies included a high proportion of female participants, with two studies focusing exclusively on breast cancer patients. The cancer types were diverse, including breast, liver, melanoma, lymphoma, sarcoma, cervical, lung, nasopharyngeal, and colon cancers. The follow-up periods ranged from 5 min to 5 days.

The pregnancy studies were conducted in Iran and Italy. The sample sizes ranged from 56 to 66 participants. The mean age of the pregnant women was in their mid-20s. The gestational age at the time of the intervention was primarily in the first trimester, averaging between 9 and 12 weeks. The follow-up periods ranged from 4 to 7 days.

#### 3.2.1. Efficacy of Peppermint Oil in Postoperative Patients

Table 1 summarises the key characteristics of the PONV studies, and Appendix A includes the extended characteristics table. We analysed the effects of peppermint oil on NV severity in postoperative patients across different time points, with a total of 453 participants at baseline (peppermint/control: 233/220), as shown in Figure 2. At 5 min after the intervention, no statistically significant difference was observed in the severity of the NV between the peppermint and the control groups (MD = −1.59 scores, 95% CI: −9.29 to 6.10, *p* = 0.467). A high heterogeneity was observed (I^2^ = 95%, CI: 89–98%). During the first two hours after the intervention, the mean difference was −0.87 scores (95% CI: −3.41 to 1.67, *p* = 0.277), with higher heterogeneity (I^2^ = 80%, CI: 36–94%). The most notable improvement occurred during the 2–6 h period, where peppermint oil showed a statistically significant benefit (MD = −0.60 scores, 95% CI: −0.77 to −0.44, *p* = 0.004), with no significant heterogeneity (I^2^ = 0%, CI: 0–90%). During the 6–12 h period, the effect size increased (MD = −0.82 scores, 95% CI: −2.65 to 1.02, *p* = 0.251), but was not statistically significant, with high heterogeneity (I^2^ = 81%, CI: 52–93%). This persisted through the 12–24 h period (MD = −0.88 scores, 95% CI: −2.29 to 0.53, I^2^ = 71%, CI: 18–90%, *p* = 0.141). By 24–48 h after the intervention, the effect decreased (MD = −0.37 scores, 95% CI: −1.61 to 0.88, *p* = 0.682), although the heterogeneity was moderate (I^2^ = 62%, CI: 0–89%). These findings suggest that peppermint oil in postoperative patients is the most effective in the 2–6 h after the intervention, with variable effects at other time points, to reduce the severity of NV. The large heterogeneity at several time points is possibly due to differences in surgical procedures, patient populations, intervention type (peppermint EO or spirit), frequency of intervention, etc.

The study by Cetin et al., 2024 [54], reported a significant difference in the incidence of PONV at baseline, with some concerns about potential bias. Specifically, significantly more patients in the control group experienced PONV immediately after surgery than the peppermint oil group (18/38 vs. 2/38, *p* = 0.001). This difference may confound the subsequent results. To address this concern, we conducted a sensitivity analysis by excluding the study by Cetin from the meta-analysis. The results of this sensitivity analysis, summarised in Appendix A, indicate that the exclusion of the study by Cetin did not substantially alter the direction of the findings at any of the time points assessed.

Appendix A shows the mean PONV scores over time in both the peppermint oil and control groups. The peppermint oil group generally showed lower NV scores than the control group, especially at later times, suggesting that peppermint oil has a potential advantage in reducing PONV. Appendix A show individual study results for the NV scores over time, comparing peppermint oil with the control. In some studies, the PONV scores were reduced with peppermint oil at later time points (24–48 h) [54,57]. However, other studies showed minimal differences [55,56]. At early time points (5–10 min), some studies [52,58] showed an initial rapid decrease in nausea. Differences between studies may be due to differences in study design, patient population, or peppermint formulation.

#### 3.2.2. Efficacy of Peppermint Oil in Pregnant Women

Table 1 summarises the key characteristics of the NV studies on pregnancy, and Appendix A provides the extended characteristics table. We analysed the effect of peppermint oil on the severity of NV in pregnant women with a total of 283 participants at baseline (peppermint/control: 189/94), as shown in Figure 3. The results varied, but the heterogeneity remained insignificant (I^2^ = 0%, CI: 0–90%) at different time points. At 24 h, the mean difference was −0.09 scores (95% CI: −0.79 to 0.61), showing no significant difference between the groups (*p* = 0.638). At 48 h, the mean difference was −0.51 scores (95% CI: −0.78 to −0.24), showing a statistically significant improvement (*p* = 0.015) in favour of the peppermint oil. At 72 h, the effect size decreased, with a mean difference of −0.20 scores (95% CI: −1.02 to 0.62) and with no statistical significance (*p* = 0.400). At 96 h, the mean difference was −0.68 scores (95% CI: −1.09 to −0.27), showing a significant improvement (*p* = 0.019) in favour of the peppermint oil. These results suggest that peppermint oil may be effective in reducing NV in pregnant women, particularly at 48 and 96 h after the intervention. Consistent, non-substantial heterogeneity indicates reliable results across studies, although effect sizes varied at different time points.

Appendix A shows pregnant women’s mean NVP scores over time (in days) in both the peppermint oil and control groups. The peppermint oil is associated with a more significant reduction in NV scores over time than the control, especially after day 2, indicating a potential benefit of peppermint oil in reducing NVP. Appendix A shows the individual study results for NVP during pregnancy, comparing peppermint oil to a control group. Although there is some variability, all three graphs show reduced NVP scores with the peppermint oil compared to the control group at seven days. The study by Pasha et al., 2012 [63], shows the largest difference between the two groups.

#### 3.2.3. Efficacy of Peppermint Oil in Chemotherapy Patients

Table 1 summarises the key characteristics of the CINV studies, and Appendix A includes the extended characteristics table. We analysed the effect of peppermint oil on the severity of NV in chemotherapy patients, with a total of 264 participants at baseline (peppermint/control: 128/136), as shown in Figure 4. The analysis demonstrated consistently favourable results for the peppermint oil at multiple time points. The difference between the effect of the peppermint oil compared to the placebo is statistically significant at 24 h after the intervention, with a mean difference of −1.85 scores (95% CI: −2.86 to −0.84, *p* = 0.004) and higher heterogeneity (I^2^ = 83%, CI: 66–91%). At 48 h, the improvement was maintained with a mean difference of −2.23 scores (95% CI: −3.13 to −1.34), statistical significance (*p* < 0.001), and moderate heterogeneity (I^2^ = 63%, CI: 16–84%). The 72 h assessment showed a consistent advantage with a mean difference of −2.41 scores (95% CI: −3.96 to −0.86), statistical significance (*p* = 0.010), and moderate heterogeneity (I^2^ = 60%, CI: 3–84%). By 96 h, while the effect size decreased slightly, it remained significant with a mean difference of −2.11 scores (95% CI: −3.48 to −0.73, *p* = 0.011) and with higher heterogeneity (I^2^ = 79%, CI: 55–91%). These results demonstrate that the peppermint oil consistently reduced the severity of NV compared to the placebo across all the analysed time points.

Appendix A shows the mean CINV scores over time (days) for cancer patients undergoing chemotherapy, comparing peppermint oil to a control group. The patients receiving the peppermint oil had lower nausea scores than the control group throughout the five-day observation period, indicating that peppermint oil has a potential benefit in reducing CINV. Appendix A shows the individual study results for CINV scores over time (in days) during chemotherapy, comparing peppermint oil to a control group. The study by Ertürk et al., 2021 [30], showed significantly lower nausea scores in the peppermint oil group using different chemotherapy regimens. It is important to note that in this study, the control group did not receive anything, which limits the ability to directly attribute the observed effects to the peppermint oil compared to the standard care or placebo. However, the studies of Jafarimanesh et al., 2020 [31], and Eghbali et al., 2017 [60], used a placebo control, and they showed smaller differences between the two groups. In most studies, there was a clear reduction in CINV scores with peppermint oil, but this varied depending on the type of chemotherapy.

#### 3.2.4. Overall Patient Satisfaction

From three postoperative studies [51,53,56] with 222 participants (peppermint/control:110/112), we analysed overall patient satisfaction at discharge, as shown in Figure 5. The mean difference between the peppermint oil and control groups was 0.68, with a 95% CI of −0.41 to 1.78. There is no statistically significant difference between the groups (*p* = 0.115). The heterogeneity was moderate, with an I^2^ value of 52% (CI: 0–86%).

Eghbali et al., 2017 [60], reported high satisfaction rates among the participants using peppermint oil (76%), with many patients recommending the treatment to others (54%). Similarly, Imani et al., 2024 [56], found significantly higher satisfaction in the peppermint group (8.77 ± 1.02) compared to the control group (7.74 ± 1.36) on a 0–10 scale (*p* < 0.001). Anderson et al., 2004 [51], observed a modest difference between the groups, with the peppermint having a slightly higher satisfaction score (86.3 ± 10.2) compared to the placebo (83.7 ± 7.4) on a 0–100 VAS scale. Notably, 93% of patients in this study would have liked to try aromatherapy in a future operation. However, Joulaeerad et al., 2018 [32], and Baek et al., 2025 [53], found no significant difference in satisfaction levels between the peppermint oil and placebo groups. These mixed results suggest that although peppermint aromatherapy may enhance patient satisfaction in some cases, its effect is inconsistent across all studies.

#### 3.2.5. Adverse Events

Appendix A summarises the reported adverse events. No adverse events were reported in the postoperative studies with the inhalation of peppermint oil [28,33,51,52,53,54,55,56,57,58,59]. In the chemotherapy studies, few patients reported adverse events. In a study by Ertürk et al., 2021 [30], 2 of 36 patients in the intervention group had headaches, and 3 experienced increased frequency and severity of nausea. In a study by Mapp et al., 2020 [62], only 1 of the 36 patients reported feeling worse after inhaling peppermint. Chemotherapy studies by Jafarimanesh et al., 2020 [31], and Eghbali et al., 2017 [60], concluded that a standard dose of peppermint oil does not cause adverse events. The study by Lestari et al., 2017 [61], did not report any adverse events after peppermint treatment. No adverse events were reported in the studies on pregnancy during the inhalation of peppermint oil [29,32,63].

#### 3.2.6. Frequency of Nausea and Vomiting

Maghami et al., 2020 [33], examined patients who underwent open-heart surgery and found significant differences between the peppermint and placebo groups in NV frequencies during the first 4 h after the intervention. The peppermint group experienced fewer nausea episodes (0.63 ± 0.81) compared to the placebo group (1.46 ± 1.21, *p* = 0.005), and fewer vomiting episodes (0.17 ± 0.46) compared to the placebo group (0.73 ± 0.60, *p* = 0.001). However, these differences were not significant in the 4–8 h and 8–12 h periods. Jafarimanesh et al., 2020 [31], found a significant difference in the reduction in vomiting frequency in the peppermint group compared to the placebo at both 24 h (0.17 ± 0.537 vs. 0.48 ± 0.707, *p* = 0.026) and 48 h (0.12 ± 0.395 vs. 0.36 ± 0.557, *p* = 0.030) after chemotherapy. However, there was no difference immediately after chemotherapy. Eghbali et al., 2017 [60], reported reductions in NV frequencies in the acute phase. For nausea, they found a significant reduction in the peppermint group (0.88 ± 0.16 standard error (SE)) compared to the control group (1.58 ± 0.21 (SE), *p* = 0.013). For vomiting, the difference was not significant (*p* = 0.15), with the peppermint group showing a frequency of 0.2 ± 0.07 (SE) compared to 0.46 ± 0.16 (SE) in the control group. They found no statistically significant difference between the groups in the delayed phase for either nausea or vomiting.

#### 3.2.7. Duration of Nausea

Maghami et al., 2020 [33], found a significant difference in the duration of nausea between the peppermint (3.78 ± 5.09 min) and the control (7.97 ± 5.55 min) groups in the first 4 h after the intervention (*p* = 0.005). This indicates that the mean duration of nausea was 2.32 times longer in the placebo group than in the intervention group. However, the differences were not statistically significant in the subsequent 8 h. Eghbali et al., 2017 [60], also reported a significant difference (*p* = 0.003) in the duration of nausea between the peppermint group (1.3 ± 0.18) and the placebo group (2.16 ± 0.22). These findings suggest that peppermint aromatherapy may effectively reduce the duration of nausea, particularly in the hours immediately following intervention.

#### 3.2.8. Requested Rescue Antiemetics

Aydin et al., 2018 [52], reported a significant reduction in the need for postoperative antiemetics, with only 3.7% in the peppermint group (1/27 patients) requiring antiemetics compared to 24.1% in the control group (7/29 patients). Anderson et al., 2004 [51], found that 60% in the peppermint group (6/10 patients) required intravenous antiemetics during recovery before inhalation, compared to 50% in the saline group (6/12 patients). At 2 min after the intervention, fewer patients requested rescue antiemetics in both groups: in the peppermint group, 20% (2/10 patients), and in the placebo group, 8% (1/12 patients). Baek et al., 2025 [53], observed a significant reduction in antiemetic use in the peppermint group (0.3 ± 1.8 mg) compared to the placebo group (4.4 ± 8.5 mg, *p* = 0.006). Cetin et al., 2024 [54], reported that no one from the peppermint group required antiemetics (0/38 patients), compared to 18.4% (7/38 patients) in the placebo group (*p* < 0.05). Sites et al., 2014 [3], found no significant difference in antiemetic efficacy between the peppermint (57.7%, 15/26 patients) and placebo (62.5%, 10/16 patients) groups (*p* = 0.76). Tate et al., 1997 [59], reported that the peppermint group (*n* = 6) required antiemetics 116 times, compared to 188 times in the placebo group (*n* = 6), indicating a reduction in antiemetic use with peppermint aromatherapy.

### 3.3. Risk of Bias Assessment and Quality of Evidence

The risk of bias was a concern across all the studies. While some studies were well-conducted, most had issues related to blinding participants and personnel because of peppermint oil’s distinctive scent. Only a few postoperative studies achieved a low risk of bias across all domains, and the pregnancy studies also had some concerns. In the chemotherapy studies, intervention adherence and reporting were generally robust, but uncertainties in randomisation remained. A summary of the risk of bias assessment is presented in Appendix A.

The GRADE assessment further shows that the certainty of evidence for peppermint oil inhalation in these settings was consistently low, mainly due to the risk of bias and imprecision from small sample sizes. Although different measurement tools contributed to some heterogeneity, inconsistency and indirectness were not major concerns. Overall, while peppermint oil inhalation may help reduce nausea and vomiting, confidence in these findings is limited. A summary of the GRADE findings is presented in Table 2.

These limitations highlight the need for cautious interpretation of the pooled results and underscore the importance of rigorous study design, including improved randomisation, effective blinding, and comprehensive reporting. Larger, well-designed trials with standardised outcome measures and better blinding are needed in future studies.

### 3.4. Publication Bias and Heterogeneity

Egger’s test for publication bias could not be performed as no outcome had enough studies (at least 10) to meet the test requirements. We found no evidence of publication bias for the outcomes. However, our analysis was underpowered due to the small number of studies.

## 4. Discussion

This meta-analysis is the first to comprehensively evaluate the efficacy of peppermint oil alone compared to a control in three distinct patient populations—PONV, CINV, and NVP—with each group analysed independently, at multiple time points. As a result, peppermint oil inhalation may be a promising complementary approach in managing NV. There was a significant difference between the peppermint and the control groups in the reduction in PONV within 2–6 h, NVP within 48 and 96 h, and CINV from 24 to 96 h, supporting the benefits of peppermint oil inhalation. However, it is important to emphasise that these findings should be considered exploratory, given the substantial methodological limitations and heterogeneity across the included studies. At all outcomes, the certainty of evidence was rated as low, primarily due to methodological concerns, such as the difficulty in blinding (given peppermint’s distinctive scent), inconsistency in measurement tools, and imprecision resulting from small sample sizes (Table 2). Nearly all the included trials had some risk of bias, with many being open-label or lacking adequate blinding. The considerable heterogeneity between the studies, such as differences in the form of peppermint (essential oil vs. spirit), dosage, and frequency of inhalation, further complicates the interpretation of the pooled results. The high I^2^ values in several analyses indicate divergent results among the studies.

An additional challenge in interpreting these findings arises from the diversity of the control interventions used across the included RCTs. While some studies employed true placebos (such as saline or water inhalation), others used active comparators, including alternative essential oils or non-olfactory interventions like controlled breathing exercises. This variability makes it difficult to isolate the specific pharmacologic effect of peppermint oil from the general therapeutic context or nonspecific effects. Additionally, as most included studies lacked blinding, patient awareness of receiving a fragrant intervention could enhance placebo responses or create positive expectations of symptom relief.

The analysis of postoperative RCTs indicated significant reductions in nausea severity following peppermint oil inhalation [28,33,52,57], while other studies reported only mild or non-significant effects but still suggested potential benefits [51,55,58]. While peppermint oil inhalation was associated with statistically significant reductions in PONV severity, the improvement was modest and generally below the established minimal clinically important difference (MCID) of approximately 1.5 points on a 0–10 nausea scale [64]. The clinical relevance of such modest improvements is uncertain. Where conventional antiemetics are unsuitable, even small symptom relief may be valued by some patients, especially given peppermint oil’s favourable safety profile, low cost, and ease of administration. This hypothesis warrants further research to determine its practical significance in clinical care. Sites et al. (2014) [58] compared controlled breathing (CB) with peppermint spirit to CB alone in PONV management. After 10 min, there was no significant difference in effectiveness (*p* = 0.61) or antiemetic rescue medication use (*p* = 0.76) between the CB alone and the CB with the peppermint spirit. The study recommended CB as a first-line treatment while acknowledging the potential benefits of peppermint spirit. Tate et al. (1997) [59] compared peppermint oil, peppermint essence, and no treatment for PONV. By day 2, the peppermint oil group had zero nausea (0.00), compared to the peppermint essence (0.05) and the placebo (0.32). The patients in the peppermint oil group also required slightly less antiemetics.

Even with multiple antiemetic drugs, PONV remains common. The Fourth Consensus Guidelines note that many patients still experience symptoms despite combination prophylaxis, highlighting the need for effective non-pharmacological options. Non-pharmacological methods, such as peppermint oil aromatherapy, may constitute a promising alternative or supplemental therapy, especially for high-risk patients or those resistant to pharmacological prophylaxis [65].

A previous meta-analysis by Wang et al. (2024) [66] evaluated the effects of ginger, lavender, and peppermint aromatherapy on PONV. In their analysis, the most significant effect was observed with ginger. Peppermint oil and lavender oil also showed significant effects. In another meta-analysis by Hines et al. (2018) [67] on aromatherapy, peppermint oil did not affect the severity of nausea after treatment (four studies, intervention/control: 68/47, 5 min). These findings suggest that peppermint oil may be beneficial for managing PONV.

The analysis of the RCTs in pregnant women indicated that inhalation of peppermint oil can effectively reduce symptoms at specific time points, particularly at 48 and 96 h. The peppermint intervention demonstrated a statistically significant reduction in NV severity compared to the placebo group [29,32]. These findings suggest that peppermint oil may be a valuable non-pharmacological option for pregnant women experiencing NVP, especially given the limitations in treatment choices due to safety concerns [68]. Despite achieving statistical significance in NVP severity with peppermint oil inhalation in pregnant women, the magnitude of improvement was modest and generally below the MCID [64]. Still, the complementary use of peppermint oil is also relevant for pregnant women, especially when standard antiemetic medications are not suitable. In these circumstances, even small improvements in symptoms may be meaningful. Two reviews mentioned the efficacy of peppermint oil on NVP. These studies are included in our analysis [69,70]. Due to the paucity of studies, ours is the first meta-analysis on peppermint oil in NVP.

The analysis of the RCTs in chemotherapy patients demonstrated the potential of peppermint oil inhalation as a low-cost, non-invasive intervention for reducing the severity of CINV [71,72]. The studies showed consistent benefits across multiple days, suggesting that peppermint oil is effective in acute and delayed CINV [73,74].

A study by Mapp et al. (2020) evaluated the efficacy of a cool, damp washcloth with peppermint essential oil versus a washcloth alone in managing CINV [62]. After 30 min, the peppermint group reported significantly better improvement than the control group using the Baxter Retching Faces scale (*p* = 0.020) [75]. Lestari et al., 2017, compared peppermint oil to standard hospital care based on the Rhodes Index [61]. After 5 min, the peppermint aromatherapy reduced CINV symptoms more significantly than the standard care (*p* = 0.001 vs. *p* = 0.02).

Standard pharmacological antiemetic prophylaxis often comes with adverse events. Commonly used agents, such as ondansetron, can cause arrhythmias [76], dexamethasone is associated with hypokalemia [77], and metoclopramide may induce extrapyramidal symptoms [78]. These adverse effects can limit the use of pharmacological antiemetics, particularly in patients with contraindications or those at higher risk for complications. In contrast, peppermint oil inhalation has a favourable safety profile and is generally well tolerated, without serious adverse events.

In a previous meta-analysis, Ahn et al. (2024) [73] examined the efficacy of essential oils, including lavender, chamomile, peppermint, orange, ginger, damask rose, and sage. Peppermint oil was the most effective in reducing nausea and vomiting. Another meta-analysis by Toniolo et al. (2021) [74] on aromatherapy found peppermint and ginger promising in alleviating NV, while chamomile and ginger reduced nausea alone. These findings suggest that peppermint oil aromatherapy may be an effective complementary intervention for managing CINV in cancer patients undergoing chemotherapy.

### 4.1. Strengths and Limitations

One of the key strengths of our study is that it provides a comprehensive and systematic review of the literature on the effects of peppermint oil inhalation on NV symptoms across diverse patient populations, including pregnant women, postoperative patients, and chemotherapy patients. The inclusion of only RCTs strengthens the reliability of our findings. In addition, the meta-analysis was restricted to RCTs that used peppermint oil as a single treatment, minimising confounding factors from additional treatments.

Despite these strengths, it is important to recognise some limitations. A major limitation is that few clinical trials, with small numbers of patients, investigated the effectiveness of peppermint oil inhalation for NV, which introduces imprecision and limits the certainty of the findings. The use of different measurement tools to assess the effect of the intervention, while allowing for scale conversion in analysis, introduced heterogeneity and inconsistency. Variations in peppermint oil interventions, including differences in dosage and, in some cases, type (peppermint oil or spirit), may also affect the interpretation. Furthermore, studies used different types of control interventions, including placebo (normal saline and distilled water), standard care (routine antiemetics and routine nursing care), and other comparators (alternative oils and controlled breathing). This heterogeneity may lead to potential confounding, as patients receiving standard care may have had access to additional supportive therapies not available in placebo-controlled trials. Some trials used inert placebos, while others used no controls. The Sites et al. (2014) study [58], for example, found no significant difference in effectiveness between controlled breathing alone and controlled breathing with peppermint spirit for postoperative nausea and vomiting, suggesting that much of the observed benefit may be attributable to nonspecific effects, such as the act of inhalation or relaxation, rather than a direct pharmacologic effect of peppermint oil. The limited number of studies in each control category prevented meaningful subgroup analysis, further restricting our ability to distinguish between specific and nonspecific effects. In addition, the risk of bias was a concern in all studies, primarily due to the lack of blinding of the participants and personnel caused by the recognisable scent of peppermint oil. Furthermore, the open-label nature of most aromatherapy studies makes it possible that some observed benefits are due to placebo or expectation effects. Since the patients might be aware that they are receiving a fragrant intervention, part of the effect may reflect relaxation or positive expectations rather than a direct pharmacologic action from the peppermint oil. Additionally, the act of inhalation itself can provide symptom relief, regardless of the aroma, making it unclear how much benefit is due to the peppermint oil’s pharmacologic action versus nonspecific effects. There is currently no direct empirical evidence supporting the clinical value of sub-MCID improvements in PONV, NVP, or CINV. Therefore, any claims about the clinical meaningfulness of such changes should be considered as hypotheses and areas for future research. A further significant limitation is the lack of systematic safety reporting in the included studies. Most trials did not provide detailed information on adverse events or tolerability. Future studies should systematically collect and report safety data. The majority of the included studies were conducted in Iran, Turkey, and the USA, which may introduce geographic and demographic biases. Differences in cultural practices, healthcare infrastructure, and patient populations could affect the applicability of our results to other settings. Future research should include more diverse populations to enhance generalizability. The retrospective nature of meta-analyses, including the present study, introduces certain limitations. We are dependent on the quality and reporting of the included RCTs, and unmeasured confounders may remain present. Moreover, publication bias and selective reporting cannot be entirely excluded. The pooled effect estimates could be overestimated due to systematic biases and the limitations described above. We interpret our findings with caution and recommend that future research prioritise rigorous study design, adequate blinding, and standardised outcome measures to improve the reliability of the evidence in this field. Despite these limitations, our study provides valuable insights into the potential efficacy of peppermint oil inhalation as a basis for future research.

### 4.2. Implications for Practice and Research

Translational science bridges the gap between clinical research and real-world practice [79,80]. The findings of this study suggest that the inhalation of peppermint oil may be an effective alternative treatment for alleviating NV symptoms in diverse patient populations. This natural and accessible intervention offers a potentially safer and a more cost-effective option compared to traditional antiemetic medications, particularly for pregnant women and those seeking non-pharmacological approaches [71,72]. The minimal adverse events associated with peppermint oil inhalation further support its consideration as a complementary therapy. However, given the low certainty of evidence, as assessed by GRADEs, these findings should be implemented with caution.

In PONV, the analysis points to a short-term benefit (2–6 h), but the strength of this evidence is limited by methodological concerns and imprecision. Future research should investigate the long-term effects of peppermint oil inhalation and its potential benefits in managing the symptoms. Larger, well-designed clinical trials with extended follow-up durations are needed to explore the optimal application of the intervention. Future studies should also focus on determining peppermint oil inhalation’s optimal dosage, frequency, and duration to maximise its therapeutic effects.

In CINV, peppermint oil inhalation is a promising complementary intervention to reduce the severity of nausea across several days of follow-up. Healthcare providers should consider including peppermint oil inhalation in supportive care plans for chemotherapy patients. However, the observed effectiveness may vary depending on the specific chemotherapeutic agent or the type of cancer. Nevertheless, clinicians should be aware of the current limitations of the evidence base, and observed effectiveness may vary depending on the specific chemotherapeutic agent or cancer type. Further research is needed to identify optimal treatment protocols.

In NVP, the inhalation of peppermint oil demonstrates potential as a non-pharmacological management option. However, the evidence is especially uncertain due to the small number of studies and patients. Larger studies with measurements at multiple time points and dosages are needed to better understand the impact of the intervention at different stages of pregnancy. Further evaluation of the impact of peppermint oil inhalation on patient quality of life and satisfaction could also be explored.

## 5. Conclusions

This systematic review and meta-analysis suggests that peppermint oil inhalation may offer a potential complementary approach for managing nausea and vomiting. However, the certainty of the evidence is low due to methodological limitations, inconsistency, and imprecision. In postoperative patients, peppermint oil inhalation appeared most effective in reducing nausea and vomiting severity within 2–6 h after the intervention, with variable effects at other time points. In pregnant women, peppermint oil inhalation was associated with reduced severity of nausea and vomiting at 48 and 96 h after the intervention, though the evidence in this group is limited by small sample sizes and study heterogeneity. In chemotherapy patients, peppermint oil inhalation consistently reduced the severity of nausea and vomiting at all analysed time points, with the most notable effects observed at 48 and 72 h, with sustained benefit through 96 h. Nevertheless, these findings are based on low-certainty evidence. While some analyses demonstrated statistically significant reductions in symptom severity, these improvements did not reach thresholds considered clinically meaningful for patients with PONV and NVP. The effects may be more promising for CINV patients, but further research is needed to clarify the clinical value of these findings. Given its favourable safety profile and accessibility, peppermint oil inhalation could be considered as an adjunct when conventional antiemetic options are limited or unsuitable. Further well-designed trials are needed to confirm the efficacy of peppermint oil and optimise its use in various clinical settings.

## Figures and Tables

**Figure 1 jcm-14-05069-f001:**
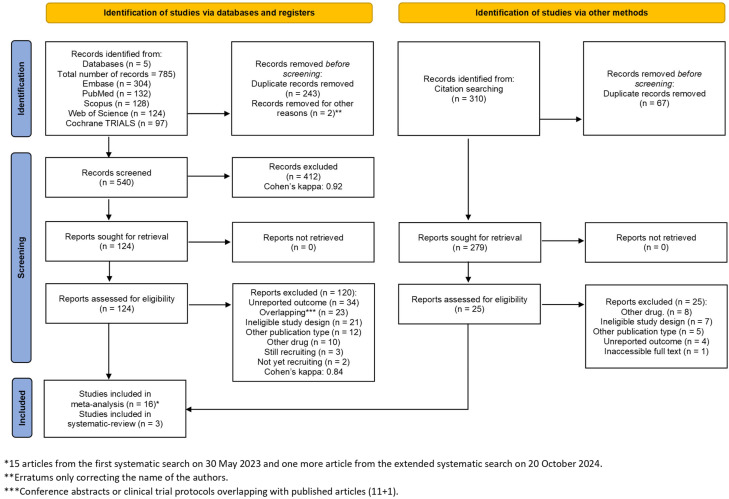
PRISMA 2020 flow diagram [36].

**Figure 2 jcm-14-05069-f002:**
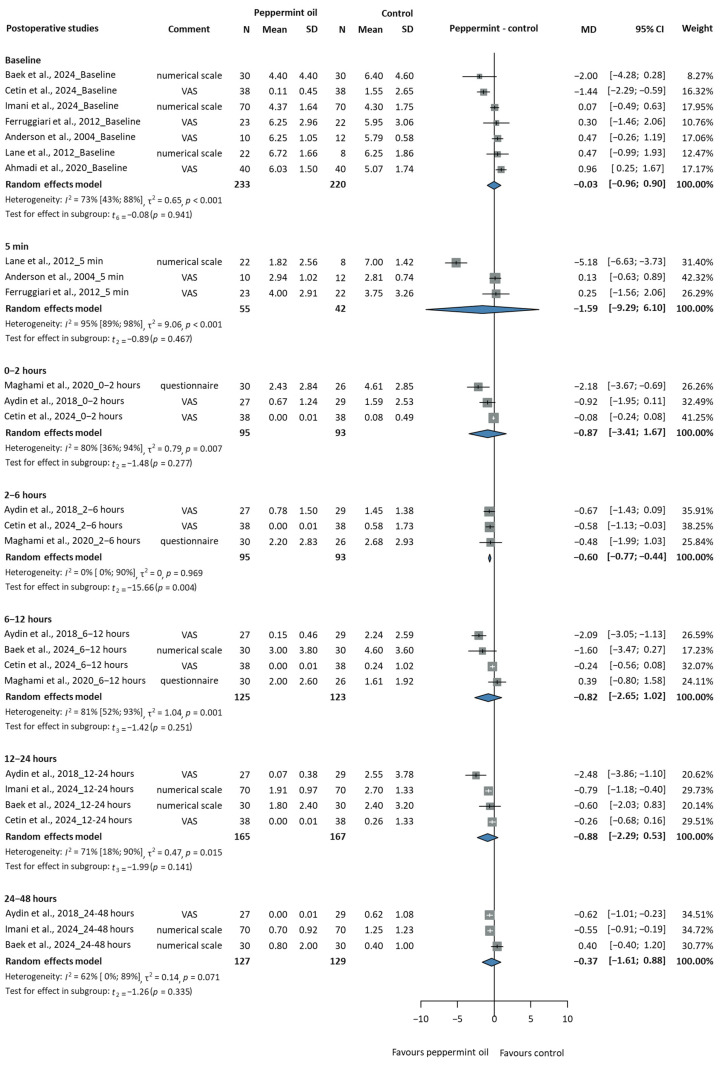
Severity of nausea and vomiting in postoperative patients (MD = mean difference, CI = confidence interval) [28,33,51,52,53,54,55,56,57].

**Figure 3 jcm-14-05069-f003:**
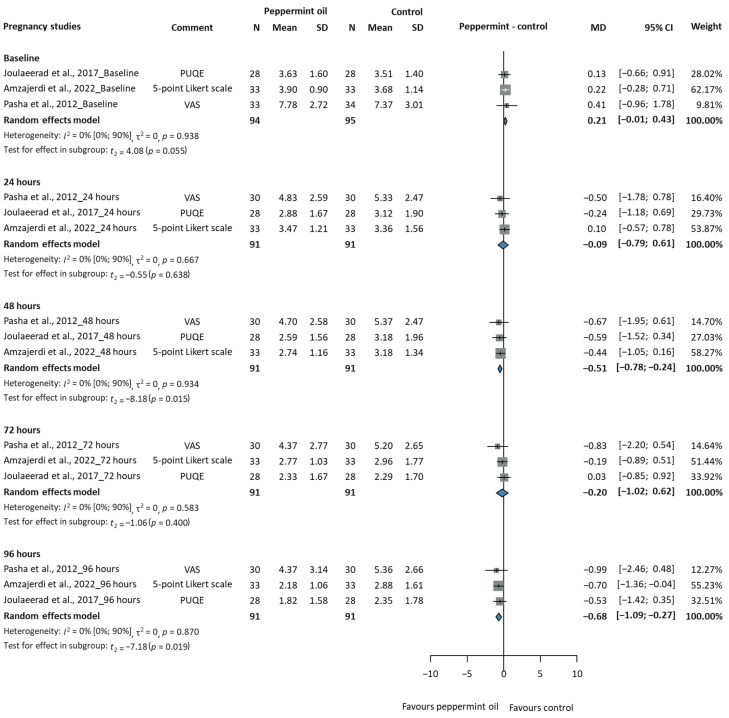
Severity of nausea and vomiting in pregnant women (MD = mean difference, CI = confidence interval) [29,32,63].

**Figure 4 jcm-14-05069-f004:**
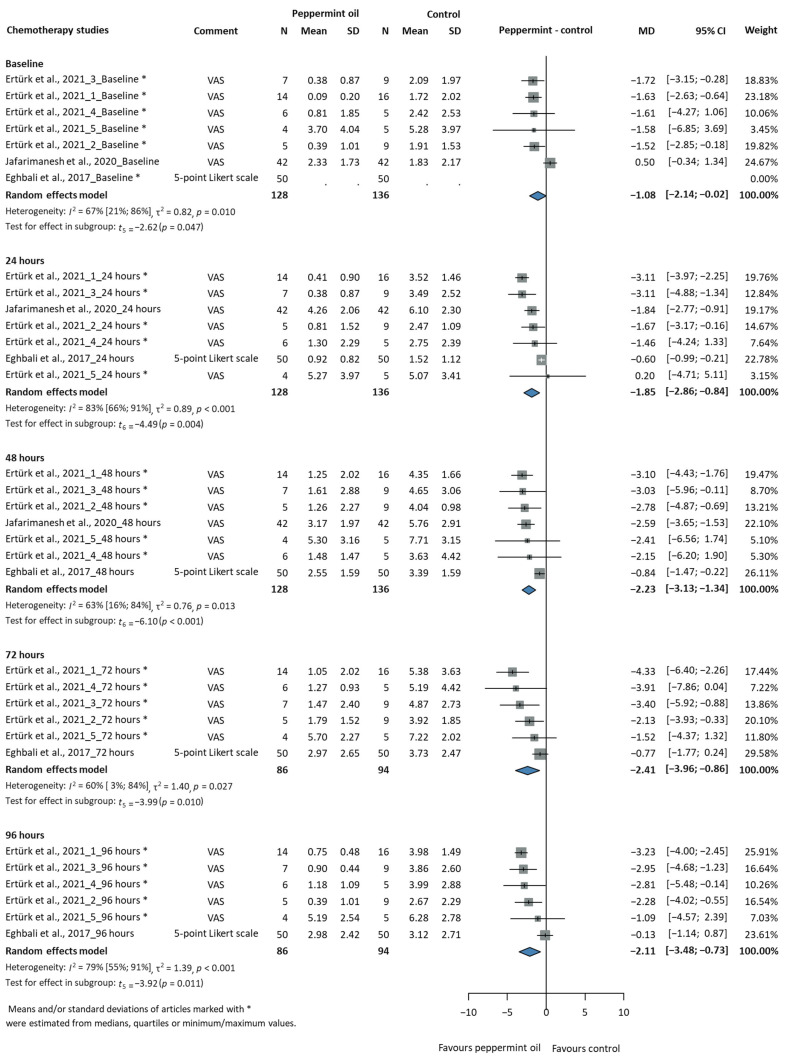
Severity of nausea and vomiting in chemotherapy patients (MD = mean difference, CI = confidence interval) [30,31,60].

**Figure 5 jcm-14-05069-f005:**
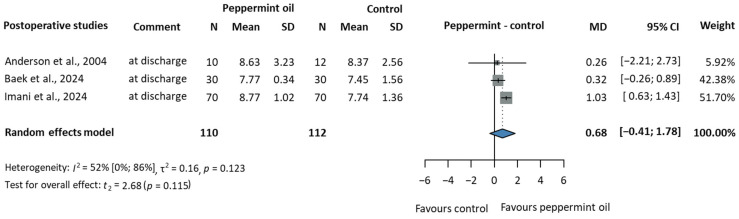
Overall satisfaction of patients with PONV at discharge after surgery (MD = mean difference, CI = confidence interval) [51,53,56].

**Table 1 jcm-14-05069-t001:** The basic characteristics of the included studies.

First Author, Publication Year	Country	Nausea Type	Surgical Procedure ^¥/^ Gestational Age ^Ꙟ^/Cancer Type ^Ꙙ^	Drug Type	Dosage	Follow-Up Period	Frequency of Intervention
Ahmadi et al., 2020 [28]	Iran	PONV	Abdominal ^¥^	Peppermint EO	2 drops (0.1 mL) of 10% EO + 2 cc DW	10 min	Once for 5 min
Maghami et al., 2020 [33]	Iran	PONV	Open-heart ^¥^	Peppermint EO	0.1 mL of EO plus 10 mL DW	0–4, 4–8, and 8–12 h	10 min before every examination
Ferruggiari et al., 2012 [55]	USA	PONV	Any ^¥^	Peppermint EO	2 drops (0.1 mL) of EO and 5 mL of 0.9% normal saline	5 and 10 min	Once for 5 min
Aydin et al., 2018 [52]	Turkey	PONV	Head, neck, eye, ear, and intraabdominal ^¥^	Peppermint EO	EO was diluted to 1/10 with wheat oil	0–2, 2–6, 6–12, 12–24, and 24–48 h	5 times in every 30 min
Lane et al., 2012 [57]	USA	PONV	Post C-section ^¥^	Peppermint spirit	1 mL peppermint spirit	2 and 5 min	At baseline, at 2 min, and at 5 min
Anderson et al., 2004 [51]	USA	PONV	Any ^¥^	Peppermint EO	0.2 mL of EO and 2 mL of isotonic saline	2 and 5 min	Once, taking three slow, deep breaths
Baek et al., 2025 [53]	South Korea	PONV	Total knee arthroplasty ^¥^	Peppermint EO	5 drops (0.25 mL) of 100% pure aroma oil	24, 48, 72 h	At least 5 nasal inspirations
Imani et al., 2024 [56]	Iran	PONV	Laparoscopic cholecystectomy ^¥^	Peppermint EO	3 drops (0.15 mL) of 100% EO	24, 48 h	5 min repeated 3 times
Cetin et al., 2024 [54]	Turkey	PONV	Cervical ^¥^	Peppermint EO	5 drops (0.25 mL) of EO	5, 35, 65, 95 min, 2, 6, 12, 24 h	Replaced every 30 min
Sites et al., 2014 [58]	USA	PONV	Laparoscopic, ear, nose, and throat, orthopaedic, or urological ^¥^	Peppermint spirit	0.5 mL peppermint spirit	5 and 10 min	3 repetitions of deep breathing
Tate et al., 1997 [59]	UK	PONV	Gynaecological ^¥^	Peppermint EO, Peppermint essence	Not defined	24, 48 h	Inhaled from the bottle when feeling nauseous
Amzajerdi et al., 2022 [29]	Iran	NVP	I: 10.31 ± 2.50 C: 10.64 ± 2.34 ^Ꙟ^	Peppermint EO	4 drops (0.2 mL) of EO diluted to 10% in sesame oil	7 days	Twice a day
Joulaeerad et al., 2017 [32]	Italy	NVP	I: 12.4 ± 3.77 C: 12.1 ± 4.06 ^Ꙟ^	Peppermint EO	5 drops (0.25 mL) of EO diluted to 10% in sweet almond oil	4 days	4 times a day
Pasha et al., 2012 [63]	Iran	NVP	I: 9.07 ± 1.31 C: 9.73 ± 2.21 ^Ꙟ^	Peppermint EO	4 drops (0.2 mL) of EO in a bowl of water	4 days	Whole night under the bed
Mapp et al., 2020 [62]	USA	CINV	any, except head and neck ^Ꙙ^	Peppermint EO	Cool damp washcloth with two drops (0.1 mL) of EO	30 min	Once
Lestari et al., 2017 [61]	Indonesia	CINV	Liver, cervical, lung, nasopharyngeal, breast, colon, melanoma, lymphoma, and sarcoma ^Ꙙ^	Peppermint EO	EO dropped onto a cotton ball	5 min	Once
Ertürk et al., 2021 [30]	Turkey	CINV	Any ^Ꙙ^	Peppermint EO, sweet almond oil	1 drop (0.05 mL) of aromatic mixture	5 days	3 times a day
Jafarimanesh et al., 2020 [31]	Iran	CINV	Breast ^Ꙙ^	Peppermint extract in tap water	40 drops (2 mL) of peppermint extract in 20 cc of tap water	1 and 2 days	Every 8 h
Eghbali et al., 2017 [60]	Iran	CINV	Breast ^Ꙙ^	Peppermint EO	2 drops (0.1 mL) of EO	1–5 days	20 min three times a day

PONV = postoperative nausea and vomiting, NVP = nausea and vomiting in pregnancy, CINV = chemotherapy-induced nausea and vomiting, EO = essential oil, I = interventional group, C = control group, mL = millilitre, cc = cubic centimetre, DW = distilled water, ^¥^ = surgical procedure, ^Ꙟ^ = gestational age, ^Ꙙ^ = cancer type

**Table 2 jcm-14-05069-t002:** Summary for quality of evidence, GRADE assessment for postoperative, pregnancy, and chemotherapy studies.

Certainty Assessment							Number of Patients		Effect		Certainty	Importance
Number of Studies	Study Design	Risk of Bias	Inconsistency	Indirectness	Imprecision	Other Considerations	Peppermint Oil	Placebo	Relative (95% CI)	Absolute (95% CI)		
PONV: Severity of nausea (follow-up: 5 min; assessed with: VAS/numerical scale; scale from 0 to 10)												
3	Randomised trials	Serious ^a^	Not serious	Not serious	Serious ^b,c^	None	55	42	-	MD 1.59 lower (9.29 lower to 6.1 higher)	⨁⨁◯◯ Low ^a,b,c^	Critical
PONV: Severity of nausea (follow-up: range 0 h to 2 h; assessed with: VAS/questionnaire; scale from 0 to 10)												
3	Randomised trials	Serious ^a^	Not serious	Not serious	Serious ^b,c^	None	95	93	-	MD 0.87 SD lower (3.41 lower to 1.67 higher)	⨁⨁◯◯ Low ^a,b,c^	Critical
PONV: Severity of nausea (follow-up: range 2 h to 6 h; assessed with: VAS/questionnaire; scale from 0 to 10)												
3	Randomised trials	Serious ^a^	Not serious	Not serious	Serious ^b,c^	None	95	93	-	MD 0.6 lower (0.77 lower to 0.44 lower)	⨁⨁◯◯ Low ^a,b,c^	Critical
PONV: Severity of nausea (follow-up: range 6 h to 12 h; assessed with: VAS/numerical scale/questionnaire; scale from 0 to 10)												
4	Randomised trials	Serious ^a^	Not serious	Not serious	Serious ^b,c^	None	125	123	-	MD 0.82 lower (2.65 lower to 1.02 higher)	⨁⨁◯◯ Low ^a,b,c^	Critical
PONV: Severity of nausea (follow-up: range 12 h to 24 h; assessed with: VAS/numerical scale; scale from 0 to 10)												
4	Randomised trials	Serious ^a^	Not serious	Not serious	Serious ^d^	None	165	167	-	MD 0.88 lower (2.29 lower to 0.53 higher)	⨁⨁◯◯ Low ^a,d^	Critical
PONV: Severity of nausea (follow-up: range 24 h to 48 h; assessed with: VAS/numerical scale; scale from 0 to 10)												
3	Randomised trials	Serious ^a^	Not serious	Not serious	Serious ^d^	None	127	129	-	MD 0.37 lower (1.61 lower to 0.88 higher)	⨁⨁◯◯ Low ^a,d^	Critical
PONV: Overall patient satisfaction (follow-up: at discharge; scale from 0 to 10)												
3	Randomised trials	Serious ^a^	Not serious	Not serious	Serious ^d^	None	110	112	-	MD 0.68 higher (0.41 lower to 1.78 higher)	⨁⨁◯◯ Low ^a,d^	Important
NVP: Severity of nausea and vomiting (follow-up: 24 h; assessed with: VAS/Likert scale/PUQE; scale from 0 to 10)												
3	Randomised trials	Serious ^a^	Not serious	Not serious	Serious ^b,c^	None	91	91	-	MD 0.09 lower (0.79 lower to 0.61 higher)	⨁⨁◯◯ Low ^a,b,c^	Critical
NVP: Severity of nausea and vomiting (follow-up: 48 h; assessed with: VAS/Likert scale/PUQE; scale from 0 to 10)												
3	Randomised trials	Serious ^a^	Not serious	Not serious	Serious ^b,c^	None	91	91	-	MD 0.51 lower (0.78 lower to 0.24 lower)	⨁⨁◯◯ Low ^a,b,c^	Critical
NVP: Severity of nausea and vomiting (follow-up: 72 h; assessed with: VAS/Likert scale/PUQE; scale from 0 to 10)												
3	Randomised trials	Serious ^a^	Not serious	Not serious	Serious ^b,c^	None	91	91	-	MD 0.2 lower (1.02 lower to 0.62 higher)	⨁⨁◯◯ Low ^a,b,c^	Critical
NVP: Severity of nausea and vomiting (follow-up: 96 h; assessed with: VAS/Likert scale/PUQE; scale from 0 to 10)												
3	randomised trials	serious ^a^	not serious	not serious	serious ^b,c^	none	91	91	-	MD 0.68 lower (1.09 lower to 0.27 lower)	⨁⨁◯◯ Low ^a,b,c^	Critical
CINV: Severity of nausea and vomiting (follow-up: 24 h; assessed with: VAS/Likert scale; scale from 0 to 10)												
3 *	Randomised trials	Serious ^a^	Not serious	Not serious	Serious ^e,f^	None	128	136	-	MD 1.85 lower (2.86 lower to 0.84 lower)	⨁⨁◯◯ Low ^a,b,c^	Critical
CINV: Severity of nausea and vomiting (follow-up: 48 h; assessed with: VAS/Likert scale; scale from 0 to 10)												
3 *	Randomised trials	Serious ^a^	Not serious	Not serious	Serious ^e,f^	None	128	136	-	MD 2.23 lower (3.13 lower to 1.34 lower)	⨁⨁◯◯ Low ^a,b,c^	Critical
CINV: Severity of nausea and vomiting (follow-up: 72 h; assessed with: VAS/Likert scale; scale from 0 to 10)												
2 *	Randomised trials	Serious ^a^	Not serious	Not serious	Serious ^e,f^	None	86	94	-	MD 2.41 lower (3.96 lower to 0.86 lower)	⨁⨁◯◯ Low ^a,b,c^	Critical
CINV: Severity of nausea and vomiting (follow-up: 96 h; assessed with: VAS/Likert scale; scale from 0 to 10)												
2 *	Randomised trials	Serious ^a^	Not serious	Not serious	Serious ^e,f^	None	86	94	-	MD 2.11 lower (3.48 lower to 0.73 lower)	⨁⨁◯◯ Low ^a,b,c^	Critical

CI: confidence interval; MD: mean difference. Explanations: a. In the articles, blinding was questionable as peppermint has a distinctive scent. b. Different measurement tools were used in the articles. c. A low patient number in all the articles. d. A low patient number in some of the articles. e. In the Eghbali article, a different measurement tool was used compared to the other articles. f. Low patient numbers in the Ertürk subgroups. * The Ertürk study analysed subgroups with different chemotherapeutic agents, and every subgroup had an individual control group for comparison.

## Data Availability

No new data were created or analysed in this study. Data sharing is not applicable to this article.

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
