# Peer review of "Inhaling Peppermint Essential Oil as a Promising Complementary Therapy in the Treatment of Nausea and Vomiting"

_jcm, 2025, doi:10.3390/jcm14145069_

Round 1

Reviewer 1 Report (Previous Reviewer 1)

Comments and Suggestions for Authors

Thank you so much for your invitation to review this manucript again. This systematic review and meta analysis evaluated the efficacy of peppermint oil inhalation for postoperative (PONV),  chemotherapy-induced (CINV), and pregnancy-related nausea and vomiting (NVP). The study follows PRISMA guidelines, includes a substantial number of RCTs (n=19), and provides a nuanced breakdown of effects by population and time interval. However there are several aspects need to be improved. I appreciate the authors’ thoughtful and thorough responses to the previous comments. The authors have strengthened the discussion of methodological limitations, especially around sample size and generalizability. The revised manuscript demonstrates improved readability. I have no further major concerns.

Author Response

1. Summary

Thank you very much for taking the time to review this manuscript again. Please find the response below.

2. Point-by-point response to Comments and Suggestions for Authors

Comments 1: Thank you so much for your invitation to review this manucript again. This systematic review and meta analysis evaluated the efficacy of peppermint oil inhalation for postoperative (PONV),  chemotherapy-induced (CINV), and pregnancy-related nausea and vomiting (NVP). The study follows PRISMA guidelines, includes a substantial number of RCTs (n=19), and provides a nuanced breakdown of effects by population and time interval. However there are several aspects need to be improved. I appreciate the authors’ thoughtful and thorough responses to the previous comments. The authors have strengthened the discussion of methodological limitations, especially around sample size and generalizability. The revised manuscript demonstrates improved readability. I have no further major concerns.

Response 1: We sincerely thank you for your thoughtful and constructive feedback throughout the review process. We greatly appreciate the positive assessment of our revised manuscript. We are pleased to hear that the revisions have enhanced the readability and overall quality of the manuscript.

Thank you again for your time and valuable input, which have significantly contributed to strengthening our work.

Reviewer 2 Report (New Reviewer)

Comments and Suggestions for Authors

General comment:

This is an interesting article investigating a real-life related topic. It will help in general population a lot! I have only some minimal comments below

  1. Since the rating scales of severity of nausea and vomiting were different across trials in Figure 2, 3, 4, and 5, I would recommend using standardized mean differences to rate the effect size but not mean differences.
  2. I recognized that it would be difficult to define the dosage of peppermint oil. However, if the authors could try to arrange subgroup based on dosage of peppermint oil, it would provide extra information to readers.
  3. Following comment 2, if possible, it would be helpful to arrange subgroup of treatment duration.

Author Response

1. Summary

Thank you very much for taking the time to review this manuscript. We greatly appreciate your careful evaluation and thoughtful consideration of our work. Please find our detailed responses to your feedback below. We highlighted sections in the resubmitted manuscript to support our responses.

2. Point-by-point response to Comments and Suggestions for Authors

Comments 1: Since the rating scales of severity of nausea and vomiting were different across trials in Figure 2, 3, 4, and 5, I would recommend using standardized mean differences to rate the effect size but not mean differences.

Response 1: Thank you for your comment regarding the use of standardized mean differences (SMD). As described in our methods, although the original studies used different scales to assess nausea and vomiting severity, we converted all outcomes to a uniform 0–10 scale before analysis. This allowed us to use mean differences (MD) as our effect size measure.

We believe using MD in this context is appropriate because all data were harmonized to the same scale, making the results directly interpretable as changes on a 0–10 scale. In contrast, SMD is mainly recommended when outcomes cannot be converted to a common metric, as it standardizes by the pooled standard deviation. However, SMD can introduce bias when standard deviations vary across studies, which is less of a concern with MD on a unified scale.

Our approach is consistent with methodological guidelines, such as those in the Cochrane Handbook, which recommend MD when possible for clearer and more clinically meaningful results.

Comments 2: I recognized that it would be difficult to define the dosage of peppermint oil. However, if the authors could try to arrange subgroup based on dosage of peppermint oil, it would provide extra information to readers.

Response 2: Thank you for your suggestion regarding subgroup analysis based on the dosage of peppermint oil. We agree that examining the effects of different dosages would provide valuable additional information for readers. However, as we have already conducted subgroup analyses with postoperative, chemotherapy-induced, and pregnancy-related nausea and vomiting—the number of studies within each subgroup is limited. Because of the small amount of data, dividing these groups by dosage would not be statistically valid.

We have acknowledged this limitation in our manuscript before (page 20, lines 567-570) and recommended that future research should focus on evaluating various dosages of peppermint oil to better inform clinical practice (page 21, lines 611-613; page 21, lines 620-621; page 21 lines 624-626).

Comments 3: Following comment 2, if possible, it would be helpful to arrange subgroup of treatment duration.

Response 3: Thank you for your suggestion to conduct a subgroup analysis based on treatment duration. We agree that exploring the impact of treatment duration would provide valuable insights. However, similar to the issue with dosage, the number of available studies within each of our main subgroups (PONV, CINV, NVP) was not sufficient to allow for further meaningful subgroup analyses according to treatment duration. We have acknowledged this limitation in our manuscript (page 20, lines 567-573) and recommend that future research include more detailed reporting on treatment duration to enable such analyses (page 21, lines 607-627). We appreciate your thoughtful comment and recognize the importance of this aspect for future studies.

Reviewer 3 Report (New Reviewer)

Comments and Suggestions for Authors

Dear Authors 

Congratulations on a very decent meta-analysis. 

Neverthelass, I would have some suggestions. 

line 50: PONV is an adverse event rather than side effect. Side effect is rather an additional effect, positive or negative whereas an adverse event contitutes an event that incidence should be prevented or requiring 

in patients undergoing vitreoretinal surgeries with diabetes PONV may result in impairment of vision 

Chandra A, Xing W, Kadhim MR, et al. Suprachoroidal hemorrhage in pars plana vitrectomy: risk factors and outcomes over 10 years. Ophthalmology 2014; 121: 311–317.

line 51-52: 

high-risk factors such as older age, co-morbidity, obesity, and alcohol consumption, as well as smoking, medications [2, 3] - that is not true 

in fact: risk factors of incicence of PONV are: female gender, non-smoking status, motion sickness, history of PONV, use of opioid analgesics 

Apfel, C.C.; Läärä, E.; Koivuranta, M.; Greim, C.A.; Roewer, N. A simplified risk score for predicting postoperative nausea and
vomiting: Conclusions from cross-validations between two centers. Anesthesiology 1999, 91, 693–700. [CrossRef]

smoking status reduces PONV 

Cohen, M.M.; Duncan, P.G.; DeBoer, D.P.; Tweed, W.A. The postoperative interview: Assessing risk factors for nausea and vomiting. Anesth. Analg. 1994, 78, 7–16. [CrossRef] [PubMed]
Stadler, M.; Bardiau, F.; Seidel, L.; Albert, A.; Boogaerts, J.G. Difference in Risk Factors for Postoperative Nausea and Vomiting. Anesthesiology 2003, 98, 46–52. [CrossRef] [PubMed] is 

obesity is controversial for development of PONV, more studies suggest less incidence of PONV in obese patients, more frequent incidence of PONV in patients with BMI <19. 

Kim JH, Hong M, Kim YJ, Lee HS, Kwon YS, Lee JJ. Effect of Body Mass Index on Postoperative Nausea and Vomiting: Propensity Analysis. J Clin Med. 2020 May 26;9(6):1612. doi: 10.3390/jcm9061612. PMID: 32466515; PMCID: PMC7355557.

Kranke, P.; Apfel, C.C.; Papenfuss, T.; Rauch, S.; Löbmann, U.; Rübsam, B.; Greim, C.-A.; Roewer, N. An increased body mass index is no risk factor for postoperative nausea and vomiting. A systematic review and results of original data. Acta Anesth.
Scand. 2001, 45, 160–166

Nitahara, K.; Sugi, Y.; Shono, S.; Hamada, T.; Higa, K. Risk factors for nausea and vomiting following vitrectomy in adults. Eur. J.Anesthesiol. 2007, 24, 166–170. 

line 53

and the type of surgery including choice of analgesics [4, 5].

I would suggest : 

type of surgery and anaesthesia regimen favouring regional over inhalational techniques  [Reibaldi, M.; Fallico, M.; Longo, A.; Avitabile, T.; Astuto, M.; Murabito, P.; Minardi, C.; Bonfiglio, V.; Boscia, F.; Furino, C.; et al.
Efficacy of Three Different Prophylactic Treatments for Postoperative Nausea and Vomiting after Vitrectomy: A Randomized
Clinical Trial. J. Clin. Med. 2019, 8, 391.] 

[Daccache N, Wu Y, Jeffries SD, Zako J, Harutyunyan R, Pelletier ED, Laferrière-Langlois P, Hemmerling TM. Safety and recovery profile of patients after inhalational anaesthesia versus target-controlled or manual total intravenous anaesthesia: a systematic review and meta-analysis of randomised controlled trials. Br J Anaesth. 2025 May;134(5):1474-1485. doi: 10.1016/j.bja.2025.02.007. Epub 2025 Mar 11. PMID: 40074622; PMCID: PMC12106880.] 

Rate of incidence of PONV may be further halved by employment of Surgical pleth index guidance of opioid free anaesthesia 

Feng CD, Xu Y, Chen S, Song N, Meng XW, Liu H, Ji FH, Peng K. Opioid-free anaesthesia reduces postoperative nausea and vomiting after thoracoscopic lung resection: a randomised controlled trial. Br J Anaesth. 2024 Feb;132(2):267-276. doi: 10.1016/j.bja.2023.11.008. Epub 2023 Dec 1. PMID: 38042725.

or even reduced below 10% by employment of Adequacy of Anaesthesia guidance

Pluta A, Stasiowski MJ, Lyssek-Boroń A, Król S, Krawczyk L, Niewiadomska E, Żak J, Kawka M, Dobrowolski D, Grabarek BO, Szumera I, Missir A, Rejdak R, Jałowiecki P. Adverse Events during Vitrectomy under Adequacy of Anesthesia-An Additional Report. J Clin Med. 2021 Sep 15;10(18):4172. doi: 10.3390/jcm10184172. PMID: 34575281; PMCID: PMC8468095.

Discussion is detailed and very interesting, but authors should also direct readers; attention to one important aspect. Antiemetic prophylaxis is not free from potential side effects: 

  • ondansetron - arrhythmia
  • dexamethasone - hypocalemia
  • metoclopramid - extrapyramidal symptoms that peppermint is free from.

Additionally despite even several drug prophylaxis Gan TJ, Belani KG, Bergese S, Chung F, Diemunsch P, Habib AS, Jin Z, Kovac AL, Meyer TA, Urman RD, Apfel CC, Ayad S, Beagley L, Candiotti K, Englesakis M, Hedrick TL, Kranke P, Lee S, Lipman D, Minkowitz HS, Morton J, Philip BK. Fourth Consensus Guidelines for the Management of Postoperative Nausea and Vomiting. Anesth Analg. 2020 Aug;131(2):411-448. doi: 10.1213/ANE.0000000000004833. Erratum in: Anesth Analg. 2020 Nov;131(5):e241. doi: 10.1213/ANE.0000000000005245. PMID: 32467512.

PONV still exists so non-pharmacological methods may constitute a promising alternative or supplemental therapy for high risk patients, resistant to pharmacological prophlaxis. 

Good luck with your minor revision 

MI to favour PONV

Author Response

1. Summary

Thank you very much for taking the time to review this manuscript. We greatly appreciate your careful evaluation and thoughtful consideration of our work. Please find our detailed responses to your feedback below. We highlighted sections in the resubmitted manuscript to support our responses.

2. Point-by-point response to Comments and Suggestions for Authors

Comments 1: line 50:

PONV is an adverse event rather than side effect. Side effect is rather an additional effect, positive or negative whereas an adverse event contitutes an event that incidence should be prevented or requiring

in patients undergoing vitreoretinal surgeries with diabetes PONV may result in impairment of vision

Chandra A, Xing W, Kadhim MR, et al. Suprachoroidal hemorrhage in pars plana vitrectomy: risk factors and outcomes over 10 years. Ophthalmology 2014; 121: 311–317.

Response 1: Thank you for your insightful comment. In accordance with your suggestion, we refer now PONV as an adverse event rather than a side effect. We modified the text as: Postoperative nausea and vomiting (PONV) are common side effects adverse events of anesthesia and surgery. (page 2 line 58-59)

Additionally, we incorporated your suggested statement and reference: In particular, in patients undergoing vitreoretinal surgeries, especially those with diabetes, PONV may result in impairment of vision due to serious complications such as suprachoroidal hemorrhage. (page 2 lines 64-66).

Comments 2: line 51-52:

high-risk factors such as older age, co-morbidity, obesity, and alcohol consumption, as well as smoking, medications [2, 3] - that is not true

in fact: risk factors of incicence of PONV are: female gender, non-smoking status, motion sickness, history of PONV, use of opioid analgesics

Apfel, C.C.; Läärä, E.; Koivuranta, M.; Greim, C.A.; Roewer, N. A simplified risk score for predicting postoperative nausea and

vomiting: Conclusions from cross-validations between two centers. Anesthesiology 1999, 91, 693–700. [CrossRef]

smoking status reduces PONV

Cohen, M.M.; Duncan, P.G.; DeBoer, D.P.; Tweed, W.A. The postoperative interview: Assessing risk factors for nausea and vomiting. Anesth. Analg. 1994, 78, 7–16. [CrossRef] [PubMed]

Stadler, M.; Bardiau, F.; Seidel, L.; Albert, A.; Boogaerts, J.G. Difference in Risk Factors for Postoperative Nausea and Vomiting. Anesthesiology 2003, 98, 46–52. [CrossRef] [PubMed]

obesity is controversial for development of PONV, more studies suggest less incidence of PONV in obese patients, more frequent incidence of PONV in patients with BMI <19.

Kim JH, Hong M, Kim YJ, Lee HS, Kwon YS, Lee JJ. Effect of Body Mass Index on Postoperative Nausea and Vomiting: Propensity Analysis. J Clin Med. 2020 May 26;9(6):1612. doi: 10.3390/jcm9061612. PMID: 32466515; PMCID: PMC7355557.

Kranke, P.; Apfel, C.C.; Papenfuss, T.; Rauch, S.; Löbmann, U.; Rübsam, B.; Greim, C.-A.; Roewer, N. An increased body mass index is no risk factor for postoperative nausea and vomiting. A systematic review and results of original data. Acta Anesth.

Scand. 2001, 45, 160–166

Nitahara, K.; Sugi, Y.; Shono, S.; Hamada, T.; Higa, K. Risk factors for nausea and vomiting following vitrectomy in adults. Eur. J.Anesthesiol. 2007, 24, 166–170.

Response 2: We corrected the list of high-risk factors for PONV in the introduction. The text now reads: They occur in approximately one out of three postoperative patient and are even more frequent in those with high-risk factors, such as older age, co-morbidity, obesity, and alcohol consumption, as well as smoking, medications female gender, non-smoking status, motion sickness, history of PONV, or the use of opioid analgesics. (page 2, lines 59-62).

We have incorporated your observation regarding the risk of vision impairment in patients undergoing vitreoretinal surgeries and also clarified the discussion on obesity as a risk factor for PONV: In particular, in patients undergoing vitreoretinal surgeries, especially those with diabetes, PONV may result in impairment of vision due to serious complications such as suprachoroidal hemorrhage [9, 10]. Obesity is controversial in the development of PONV. Some studies suggest a lower incidence of PONV in obese patients, while a higher incidence is observed in patients with a BMI below 19. (page 2, lines 64-68).

We included the suggested references to support these statements.

Comments 3: line 53

and the type of surgery including choice of analgesics [4, 5].

I would suggest :

type of surgery and anaesthesia regimen favouring regional over inhalational techniques  [Reibaldi, M.; Fallico, M.; Longo, A.; Avitabile, T.; Astuto, M.; Murabito, P.; Minardi, C.; Bonfiglio, V.; Boscia, F.; Furino, C.; et al.

Efficacy of Three Different Prophylactic Treatments for Postoperative Nausea and Vomiting after Vitrectomy: A Randomized

Clinical Trial. J. Clin. Med. 2019, 8, 391.]

[Daccache N, Wu Y, Jeffries SD, Zako J, Harutyunyan R, Pelletier ED, Laferrière-Langlois P, Hemmerling TM. Safety and recovery profile of patients after inhalational anaesthesia versus target-controlled or manual total intravenous anaesthesia: a systematic review and meta-analysis of randomised controlled trials. Br J Anaesth. 2025 May;134(5):1474-1485. doi: 10.1016/j.bja.2025.02.007. Epub 2025 Mar 11. PMID: 40074622; PMCID: PMC12106880.]

Rate of incidence of PONV may be further halved by employment of Surgical pleth index guidance of opioid free anaesthesia

Feng CD, Xu Y, Chen S, Song N, Meng XW, Liu H, Ji FH, Peng K. Opioid-free anaesthesia reduces postoperative nausea and vomiting after thoracoscopic lung resection: a randomised controlled trial. Br J Anaesth. 2024 Feb;132(2):267-276. doi: 10.1016/j.bja.2023.11.008. Epub 2023 Dec 1. PMID: 38042725.

or even reduced below 10% by employment of Adequacy of Anaesthesia guidance

Pluta A, Stasiowski MJ, Lyssek-Boroń A, Król S, Krawczyk L, Niewiadomska E, Żak J, Kawka M, Dobrowolski D, Grabarek BO, Szumera I, Missir A, Rejdak R, Jałowiecki P. Adverse Events during Vitrectomy under Adequacy of Anesthesia-An Additional Report. J Clin Med. 2021 Sep 15;10(18):4172. doi: 10.3390/jcm10184172. PMID: 34575281; PMCID: PMC8468095.

Response 3: We updated the text to specify the influence of both the type of surgery and the anesthesia regimen: The type of surgery including choice of analgesics and anaesthesia regimen can influence the risk of PONV with regional techniques being preferred over inhalation techniques. (page 2, lines 62-64).

We incorporated your recommendations regarding strategies to further reduce PONV incidence. The text now states: The incidence of PONV may be further reduced by using the Surgical Pleth Index guidance of opioid-free anaesthesia [13], or even lowered below 10% with the Adequacy of Anaesthesia guidance. (page 2 lines 68-70).

We included the suggested references to support these statements.

Comments 4: Discussion is detailed and very interesting, but authors should also direct readers; attention to one important aspect. Antiemetic prophylaxis is not free from potential side effects:

ondansetron - arrhythmia

dexamethasone - hypocalemia

metoclopramid - extrapyramidal symptoms that peppermint is free from.

Gan TJ, Belani KG, Bergese S, Chung F, Diemunsch P, Habib AS, Jin Z, Kovac AL, Meyer TA, Urman RD, Apfel CC, Ayad S, Beagley L, Candiotti K, Englesakis M, Hedrick TL, Kranke P, Lee S, Lipman D, Minkowitz HS, Morton J, Philip BK. Fourth Consensus Guidelines for the Management of Postoperative Nausea and Vomiting. Anesth Analg. 2020 Aug;131(2):411-448. doi: 10.1213/ANE.0000000000004833. Erratum in: Anesth Analg. 2020 Nov;131(5):e241. doi: 10.1213/ANE.0000000000005245. PMID: 32467512.

Additionally despite even several drug prophylaxis, PONV still exists so non-pharmacological methods may constitute a promising alternative or supplemental therapy for high risk patients, resistant to pharmacological prophlaxis.

Response 4: Thank you for this important observation and for providing supporting references. We have revised the manuscript to emphasize the safety considerations of antiemetic prophylaxis and the potential role of non-pharmacological therapies: Standard pharmacological antiemetic prophylaxis often comes with adverse events. Commonly used agents such as ondansetron can cause arrhythmias [77], dexamethasone is associated with hypokalemia [78], and metoclopramide may induce extrapyramidal symptoms [79]. These adverse effects can limit the use of pharmacological antiemetics, particularly in patients with contraindications or those at higher risk for complications. In contrast, peppermint oil inhalation has a favorable safety profile and is generally well tolerated, without SAEs. (page 20 lines 545-551).

We further clarified the need for alternatives: Even with multiple antiemetic drugs, PONV remains common. The Fourth Consensus Guidelines note that many patients still experience symptoms despite combination prophylaxis, highlighting the need for effective non-pharmacological options. Non-pharmacological methods such as peppermint oil aromatherapy may constitute a promising alternative or supplemental therapy, especially for high-risk patients or those resistant to pharmacological prophylaxis [66]. (page 19 lines 507-512).

We included the suggested references to support these statements.

Reviewer 4 Report (New Reviewer)

Comments and Suggestions for Authors

This manuscript presents a systematic review and meta-analysis evaluating the efficacy of inhaled peppermint essential oil for nausea and vomiting (NV) in three contexts: postoperative (PONV), chemotherapy-induced (CINV), and pregnancy-related (NVP). The authors followed PRISMA guidelines and registered a protocol on PROSPERO . A comprehensive literature search (five databases) yielded 19 randomized controlled trials (RCTs) that met inclusion criteria . Separate meta-analyses were performed for each patient group. The results indicate that peppermint oil inhalation yielded statistically significant reductions in nausea severity in PONV during the early postoperative period (especially 2–6 hours post-surgery), in NVP at 48 and 96 hours of treatment, and in CINV across multiple days of chemotherapy . For example, in postoperative patients the pooled mean difference in nausea scores was about –0.60 (on a numeric scale) favoring peppermint at 2–6 hours post-intervention , and in chemotherapy patients the largest benefit was seen ~48 hours after chemo (mean difference around –2.23) . All these effects were statistically significant. The authors conclude that inhaled peppermint oil appears to be a “promising complementary therapy” for reducing NV in the postoperative, chemotherapy, and pregnancy settings . They note, however, that the certainty of evidence is low due to methodological issues (e.g. lack of blinding, heterogeneous measures) and small sample sizes in the included studies . Overall, the review addresses an interesting clinical question and suggests that peppermint aromatherapy may have some benefit for NV management in various populations, while calling for cautious interpretation given the limitations.

Major Concerns

  1. The reliability of the findings is limited by the generally low quality and consistency of the evidence. The authors themselves rated the certainty of evidence as “low” for all outcomes, primarily due to methodological concerns such as difficulty blinding participants (the aroma of peppermint is unmistakable), inconsistency in measurement tools across studies, and imprecision from small sample sizes . Indeed, nearly all included trials had some risk of bias – for example, complete blinding was often not feasible, and many studies were open-label, introducing potential expectancy effects. There was also considerable clinical and statistical heterogeneity among studies: trials differed in the form of peppermint (essential oil vs. “peppermint spirit”), dosage/frequency of inhalation, and the control conditions used . Such variations make it challenging to derive a single pooled estimate of effect. High I² values reported in several analyses (often >80% heterogeneity) further indicate divergent results between studies. This heterogeneity and risk of bias raise concerns about the strength of the conclusions – the meta-analytical findings might be driven by a few biased or context-specific results. The Discussion should emphasize more strongly that the evidence is exploratory. It may be advisable for the authors to perform sensitivity analyses (e.g. excluding high risk-of-bias studies) or at least acknowledge that the pooled effects could be overestimated due to systematic biases.
  2. While the meta-analysis reports statistically significant differences in nausea/vomiting scores, the magnitude of these improvements may be too small to be clinically meaningful. For instance, in postoperative and pregnancy-related nausea, peppermint oil only reduced symptom severity by roughly 0.5–0.7 points on the scales used . The manuscript does not discuss whether such small mean differences would be perceptible or important to patients. Prior research suggests that a change of about 15 mm on a 100 mm visual analog scale (≈1.5 points on a 0–10 scale) is the minimum clinically significant improvement in nausea . Many of the reported benefits of peppermint oil (e.g. ~0.5 point reductions in PONV or NVP severity) fall well below this threshold, calling into question their practical impact. This is a critical point that is currently missing from the Discussion. The authors should temper their conclusions by noting that although some results are statistically significant, the clinical relevance of such modest improvements is uncertain. In future revisions, the discussion could explicitly state whether the effect sizes observed are likely to translate into a noticeable relief for patients.
  3. The included RCTs employed a variety of control interventions – some used true placebos (e.g. saline or water inhalation), but others used active comparators like alternative oils or even non-olfactory interventions (e.g. controlled breathing exercises) . This variability is acknowledged as heterogeneity, but another implication is that the specific effect of peppermint oil vs. general therapeutic context is hard to isolate. In trials where the control was an active treatment (such as breathing techniques or another scent), any added benefit of peppermint was often minimal. Notably, one cited study (Sites et al. 2014) found no significant difference in PONV relief between patients performing controlled breathing with peppermint versus controlled breathing alone , suggesting that a large part of the benefit might come from the breathing/relaxation or placebo effect rather than peppermint itself. The review would be strengthened by a deeper discussion of this issue: the authors should mention that patient unblinding (due to peppermint’s distinctive smell) and the soothing ritual of inhalation could produce placebo responses. As a limitation, it should be acknowledged that it’s unclear how much of the observed benefit is a direct pharmacologic effect of peppermint oil and how much is a non-specific effect of aroma or breathing. If data allow, a subgroup analysis separating trials with inert placebo controls from those with active controls could be informative – differences in outcomes between these might indicate the degree of placebo effect. At minimum, the authors should caution readers that peppermint’s efficacy might not greatly exceed that of bland aromas or breathing exercises, given the mixed results across control conditions.
  4. The title of the manuscript – “Inhaling Peppermint Essential Oil is Beneficial in the Treatment of Nausea and Vomiting” – reads as a definitive statement. This could be seen as overstating the findings, considering that the review’s own analysis rates the evidence as low-certainty and notes multiple limitations. It is recommended to moderate the title and similar statements. For example, a more cautious title might be “…Potentially Beneficial…” or “…A Promising Complementary Therapy…”, which the authors actually use in the abstract conclusion . Similarly, in the Conclusion section of the paper and the Abstract, the language should reflect that the results suggest a potential benefit rather than an established one. The authors do note in the Discussion and Conclusions that the evidence is limited (e.g. “the certainty of evidence is low” ), but this caution is somewhat undermined by the unequivocal phrasing in the title and some summary statements. Aligning the tone throughout the manuscript to be consistently careful would enhance credibility. The journal’s readership should not be misled to believe peppermint inhalation is a proven therapy; rather, it should be presented as an interesting adjunct that shows promise but requires further high-quality confirmation.
  5. The review focuses on efficacy (reduction in nausea/vomiting scores) but provides little detail on adverse events or safety of peppermint oil inhalation. This is especially important given the patient populations (e.g. pregnant women, postoperative patients) where safety is paramount. In the Discussion, the authors briefly mention that peppermint oil is a natural therapy with minimal adverse effects and that it is considered safe in pregnancy (category B2) per one reference . However, the manuscript does not systematically report any side effects observed in the trials. Were there any adverse events or complaints (such as allergic reactions, heartburn, headaches, etc.) reported in the RCTs included? If the included studies tracked these, the review should summarize them. Even if no significant adverse outcomes were noted, stating this explicitly is valuable. Conversely, if safety data were not reported by many trials, that is a limitation to acknowledge (potential publication bias towards reporting only benefits). The authors should consider adding a brief results subsection on safety/tolerability. This would reassure readers that peppermint inhalation, as suggested, is indeed low-risk, and it would align with the claim that it’s a “safer option” for patients .

Minor Concerns

  • In the Introduction, the manuscript states that PONV occurs in about one-third of patients and “even more frequently in patients exposed to high-risk factors such as older age, co-morbidity, obesity, and alcohol consumption, as well as smoking…” . This appears to conflict with established knowledge. Typically, youngerage (not older) and non-smoking status (not smoking) are known patient-specific risk factors for PONV . In fact, smoking and older age have been associated with lower incidence of PONV in some risk scores . It seems the authors may have misinterpreted sources or made a typo in listing risk factors. They cite references [2] and [3] for these claims, but reference 3 is an RCT about peppermint aromatherapy (not a source on PONV risk factors). This should be corrected for accuracy. The authors should revise that sentence to reflect correct risk factors (e.g. younger age, female sex, non-smoker, history of PONV, use of opioids, etc.) or cite a proper reference that supports the risk factors listed.
  • A few sentences could be clarified or grammatically adjusted for better readability. For example, the Introduction contains the line: “Although several randomized controlled trials (RCTs) have been conducted to evaluate the effect of peppermint on PONV, NVP, and CINV; its clinical efficacy has not been established yet [20-25].” . The use of a semicolon after “CINV” is not standard because the clause after the semicolon is dependent on the “Although” clause. It would be smoother to split this into two sentences or use a comma: e.g. “Although several RCTs have evaluated peppermint oil for PONV, CINV, and NVP, its clinical efficacy has yet to be established [20–25].” Similarly, in section 4.2 (Implications for Practice and Research), there is a duplicated sentence about the need for larger studies in NVP: “Larger studies with measurements at multiple time points are needed to better understand the impact of the intervention at different stages of pregnancy.” – this exact sentence appears twice in succession . The authors should remove the redundant sentence. Aside from these, the overall writing is clear.
  • The text should ensure proper spacing and notation around numeric values, units, and confidence intervals. In the Abstract Results (and elsewhere), there are instances of missing spaces, for example: “(MD:-0.60points, 95%confidence interval (CI):-0.77 to -0.44, p=0.004)” . This should be formatted as “MD = –0.60 points, 95% confidence interval (CI): –0.77 to –0.44, p = 0.004.” for clarity. Similar spacing issues occur with other values (e.g., “MD:-2.23, 95%CI:-3.13 to -1.34” should have spaces after the colon). These are minor typographical details but are important for professionalism and readability. The authors should review all statistical reporting to ensure consistency with journal style (e.g. including leading zeroes, using en-dashes for negative signs consistently, etc.).
  • The review mentions “peppermint spirit” in a couple of instances when discussing prior studies . It may be unclear to readers what “peppermint spirit” refers to – presumably an ethanol-based peppermint preparation used in older studies or clinical practice. Since this term is not widely known, consider adding a brief explanation in the text or a footnote. For example, when first mentioning peppermint spirit, the authors could note it is an alcohol-based solution of peppermint oil traditionally used for nausea relief. This will help avoid confusion, as readers might think “spirit” is a typo or a completely different substance. Consistently using the term “peppermint essential oil (peppermint EO)” elsewhere but then referencing “peppermint spirit” could be clarified for consistency.
  • The manuscript references several figures and tables in the Appendix/Supplement (e.g., Figures S9–S14 for risk of bias, Tables S3–S5 for extended characteristics) which is appropriate. One minor point is to ensure that all these supplementary items are correctly labeled and cited in the text. For instance, Table 1 in the main text summarizes basic study characteristics, and the text says extended data is in Table S2 (likely a typo, since later it mentions Tables S3–S5 for subgroups) . Double-check the numbering of supplementary tables/figures and their call-outs to avoid confusion. Also, if possible, ensure that important outcomes like risk-of-bias summaries or GRADE assessments that are relegated to supplements are at least briefly described in the main text (the authors do mention them in words, which is good). No inconsistencies were noted in the results between text and tables as far as can be seen, but the authors should carefully proofread numbers (mean differences, CIs, p-values) to ensure they match between the abstract, main text, and any figures. Any minor discrepancies should be corrected.

Limitations Not Acknowledged by the Authors

  • Lack of discussion on clinical importance of effects: As noted in the major concerns, the manuscript does not acknowledge that statistically significant improvements might not equate to clinically meaningful benefits. This is a limitation of the evidence synthesis – if an intervention’s effect size is below the threshold of patient-noticeable improvement, its practical value is limited. The authors should explicitly acknowledge this point as a limitation. Currently, the discussion focuses on statistical outcomes and evidence certainty, but not on whether the degree of nausea reduction would matter to patients.
  • Unaddressed placebo and expectation effects: While the authors discuss heterogeneity due to different comparators, they stop short of explicitly noting that the open-label nature of aromatherapy trials and active control designs could mean the observed benefits are partly placebo effects. This is a potential weakness of all aromatherapy studies: patients know they are receiving a fragrant intervention, which could relax them or create positive expectation of relief. The manuscript does not directly call this out as a limitation. It would strengthen the critical appraisal if the authors mention that due to lack of blinding, some proportion of the effect could be non-specific. Acknowledging this would not diminish the findings, but would frame them appropriately (i.e. peppermint oil’s observed efficacy includes any placebo effect inherent to aromatherapy).
  • No aggregation of safety data: The review did not compile or report adverse event data from the trials, and the authors did not list this omission as a limitation. If none of the studies reported safety outcomes, that itself is a limitation to note (because it means we have limited evidence on harms). If studies did report it and the authors chose not to meta-analyze or describe it, that is an oversight. Either way, not addressing safety outcomes in the results is a gap. The authors should acknowledge that “potential side effects were not systematically analyzed” if applicable. This would inform readers that, for example, issues like allergic reactions or other side effects were outside the scope or rarely reported. Given that the context includes pregnant patients, the absence of documented safety data should be highlighted as a limitation in applying these results.
Comments on the Quality of English Language

Language and Grammar

The manuscript is generally well-written and clear. There are only a few grammatical or stylistic issues to address:

  • Improve sentences that currently read awkwardly. For instance, the use of “;” after an introductory “Although…” (Introduction, lines 75–77) is incorrect – splitting into two sentences or using a comma would fix this. Ensure dependent clauses are properly connected to main clauses. Also, check for any overly long sentences that could be broken up for clarity. Most of the text is concise, but a careful read-through could spot places to simplify phrasing.
  • Remove the repeated sentence in the Implications for Practice section regarding NVP (as noted above, the line about “Larger studies with multiple time points…” appears twice back-to-back ). Also, verify that no other passages are unintentionally duplicated.
  • The manuscript mostly uses past tense for the review’s methods and present tense for discussing results, which is appropriate. Just maintain consistency (e.g., sometimes “was” vs “is” in describing known facts can be mixed). The tone is appropriately formal and scientific. One suggestion is to ensure cautious language where needed (see major concern about not overstating findings). Phrases like “appears to” or “may” are used in several places – continue using those to avoid definitives that the data don’t support strongly.
  • Fix small typographical errors such as missing spaces (e.g., in “95%CI” as mentioned), inconsistent use of hyphens/en-dashes, or any misspelled words. We noted the typo “To What Extend” in reference 50 (should be “Extent”) – likely a typo in the reference list (see below). Such minor errors should be corrected to improve overall professionalism.

Reference and Citation Issues

  • Reference formatting consistency: The references need some editing to adhere to journal format. For example, Reference #1 (Zhong et al.) is listed as “Int J Mol Sci. 2021, 11, 22.” , which is confusing/mislabeled – International Journal of Molecular Sciences volume 22 (issue 11) in 2021 should likely be formatted as 2021, 22(11), [page or article ID]. It appears the volume/issue may be reversed or an article number is missing. Likewise, Reference #50 (Eghbali et al.) ends with “J Hematol Thromboemb Dis. 2017, 05.” – the “05” is presumably the volume number, but it should not have a leading zero and should include issue/pages or an article ID. The authors should review each reference for completeness: ensure every citation has Journal name, year, volume, and page range or e-location as required by the journal. Several references in the list have all components (e.g., most of the RCT references are complete), but a few like #1 and #50 seem incomplete or oddly formatted. This needs correction in the final bibliography.
  • Reference title typos and capitalization: In reference #50, the title is given as “To What Extend Aromatherapy with Peppermint Oil Effects on Chemotherapy Induced Nausea and Vomiting in Patient Diagnosed with Breast Cancer? A Randomized Controlled Trial.” . This looks like a typo (“Extend” should be “Extent”, “Effects” should likely be “Affects” or “Effects on…?”). It’s possible the original article had a grammatical error in the title, but more likely it’s a transcription mistake. The authors should verify the exact title of that source and correct the spelling (“Extent”) and wording if necessary. Additionally, ensure consistent capitalization in titles (most seem fine, just make sure things like acronyms or proper nouns are appropriately capitalized and others are not, according to journal style).
  • Punctuation and spacing in references: Some reference entries are missing a period at the end or have inconsistent punctuation. For instance, reference #49 ends with page “90-104” without a period in the text we see (the format should typically include a period at the end of each citation). Reference #60 lists an article ID (e118983) – which is fine – just ensure the formatting around it is consistent (it appears as 2024, 218, e118983which is correct for a volume and article number, but double-check spacing and punctuation). These are minor formatting quibbles, but since JCM/MDPI has specific reference style, it’s worth carefully proofreading the entire reference list.
  • Appropriateness of citations for statements: The majority of in-text citations appear appropriate and up-to-date. The authors have done well to include recent studies (even 2024 publications) and relevant prior reviews . However, there is one notable mismatch: in the Introduction, as mentioned, references [2] and [3] are cited for risk factors of PONV , but reference [3] is not about risk factors. It’s unusual to cite an aromatherapy RCT (Sites et al. 2014) in the middle of a sentence about “older age, co-morbidity, smoking” being risk factors. This likely indicates a citation error (perhaps the authors intended to cite a different source for PONV risk factors). The authors should correct this by citing an authoritative source (e.g., an anesthesiology review or guidelines) for PONV risk factors in that spot, instead of the RCT. Ensuring each statement is backed by a relevant citation is crucial. Another example to double-check: the statement “peppermint oil blocks serotonin receptors in the gut” is backed by reference [17] – the authors should ensure reference 17 indeed supports that mechanistic claim (assuming it’s a pharmacological study on menthol). These are minor content-to-citation alignments to verify.
  • Citation order and completeness: The reference numbering appears to be in reasonable order of appearance. We did not find any obviously uncited references in the list. One suggestion is to ensure all included RCTs in the meta-analysis are cited in the text (most are cited in the Results/Discussion when describing individual study findings, which is good). If any study was included but not mentioned, it might be useful to cite it when discussing the results for completeness. The authors might consider adding a PRISMA flow diagram (Figure 1 was likely such, as noted ) which helps in referencing how many studies were included/excluded – presumably they have that. As long as all 19 RCTs are properly referenced somewhere (which, given references like [20–25], [43–48] etc., seems to be the case), that is fine.

Author Response

1. Summary

Thank you very much for taking the time to review this manuscript. We greatly appreciate your careful evaluation and thoughtful consideration of our work. Please find our detailed responses to your feedback below. We highlighted sections in the resubmitted manuscript to support our responses.

2. Point-by-point response to Comments and Suggestions for Authors

Comments 1: The reliability of the findings is limited by the generally low quality and consistency of the evidence. The authors themselves rated the certainty of evidence as “low” for all outcomes, primarily due to methodological concerns such as difficulty blinding participants (the aroma of peppermint is unmistakable), inconsistency in measurement tools across studies, and imprecision from small sample sizes . Indeed, nearly all included trials had some risk of bias – for example, complete blinding was often not feasible, and many studies were open-label, introducing potential expectancy effects. There was also considerable clinical and statistical heterogeneity among studies: trials differed in the form of peppermint (essential oil vs. “peppermint spirit”), dosage/frequency of inhalation, and the control conditions used . Such variations make it challenging to derive a single pooled estimate of effect. High I² values reported in several analyses (often >80% heterogeneity) further indicate divergent results between studies. This heterogeneity and risk of bias raise concerns about the strength of the conclusions – the meta-analytical findings might be driven by a few biased or context-specific results. The Discussion should emphasize more strongly that the evidence is exploratory. It may be advisable for the authors to perform sensitivity analyses (e.g. excluding high risk-of-bias studies) or at least acknowledge that the pooled effects could be overestimated due to systematic biases.

Response 1: Thank you for your thoughtful and constructive feedback. We agree that the reliability of our findings is limited by the generally low quality and consistency of the available evidence. In response to your comments, we have revised the Discussion section to more strongly emphasize the exploratory nature of the evidence, the impact of risk of bias and heterogeneity, and the potential for overestimation of pooled effects. We have also added these limitations and clarified the implications for interpretation.

We updated the text as: However, it is important to emphasize that these findings should be considered exploratory, given the substantial methodological limitations and heterogeneity across the included studies. (page 18, lines 468-470)

Nearly all included trials had some risk of bias, with many being open-label or lacking adequate blinding. The considerable heterogeneity between the studies, such as differences in the form of peppermint (essential oil vs. spirit), dosage and frequency of in-halation, further complicates the interpretation of pooled results. High I² values in several analyses indicate divergent results between studies. (page 18, lines 474-478)

The pooled effect estimates could be overestimated due to systematic biases and the limitations described above. We interpret our findings with caution and recommend that future research prioritize rigorous study design, adequate blinding, and standardized outcome measures to improve the reliability of evidence in this field. (page 20, lines 590-594)

Comments 2: While the meta-analysis reports statistically significant differences in nausea/vomiting scores, the magnitude of these improvements may be too small to be clinically meaningful. For instance, in postoperative and pregnancy-related nausea, peppermint oil only reduced symptom severity by roughly 0.5–0.7 points on the scales used . The manuscript does not discuss whether such small mean differences would be perceptible or important to patients. Prior research suggests that a change of about 15 mm on a 100 mm visual analog scale (≈1.5 points on a 0–10 scale) is the minimum clinically significant improvement in nausea . Many of the reported benefits of peppermint oil (e.g. ~0.5 point reductions in PONV or NVP severity) fall well below this threshold, calling into question their practical impact. This is a critical point that is currently missing from the Discussion. The authors should temper their conclusions by noting that although some results are statistically significant, the clinical relevance of such modest improvements is uncertain. In future revisions, the discussion could explicitly state whether the effect sizes observed are likely to translate into a noticeable relief for patients.

Response 2: We appreciate the reviewer’s important observation regarding the distinction between statistical significance and clinical relevance. While the mean reductions in nausea and vomiting severity observed with peppermint oil inhalation are below the commonly cited minimal clinically important difference of about 1.5 points, it is essential to consider the clinical context. Peppermint oil is often used as a complementary therapy, or in cases when antiemetics are contraindicated, or not preferred. In such cases, even smaller symptomatic improvements may be meaningful to patients, particularly given peppermint oil’s favorable safety profile and minimal adverse effects. We have revised the Discussion to acknowledge this.

Specific revisions include:

Discusson:

While peppermint oil inhalation was associated with statistically significant reductions in PONV severity, the improvement was modest and generally below the established minimal clinically important difference (MCID) of approximately 1.5 points on a 0–10 nausea scale [65]. However, peppermint oil is primarily used as a complementary therapy when conventional antiemetics are contraindicated or not preferred. In these situations, even modest symptom relief may be valuable, especially given peppermint oil’s favorable safety profile, low cost, and ease of administration. (page 18, lines 489-498)

Despite achieving statistical significance in NVP severity with peppermint oil inhalation in pregnant women, the magnitude of improvement was modest and generally below the MCID [65]. Still, the complementary use of peppermint oil is also relevant for pregnant women, especially when standard antiemetic medications are not suitable. In these circumstances, even small improvements in symptoms may be meaningful. (page 19, lines 525-530)

Conclusion:

While some analyses demonstrated statistically significant reductions in symptom severity, these improvements did not reach thresholds considered clinically meaningful for PONV and NVP patients; however, the effects appeared more promising for CINV patients. Still, given its favorable safety profile and accessibility, peppermint oil may be considered when conventional antiemetic options are limited or unsuitable. (page 21, lines 638-654)

Comments 3: The included RCTs employed a variety of control interventions – some used true placebos (e.g. saline or water inhalation), but others used active comparators like alternative oils or even non-olfactory interventions (e.g. controlled breathing exercises) . This variability is acknowledged as heterogeneity, but another implication is that the specific effect of peppermint oil vs. general therapeutic context is hard to isolate. In trials where the control was an active treatment (such as breathing techniques or another scent), any added benefit of peppermint was often minimal. Notably, one cited study (Sites et al. 2014) found no significant difference in PONV relief between patients performing controlled breathing with peppermint versus controlled breathing alone , suggesting that a large part of the benefit might come from the breathing/relaxation or placebo effect rather than peppermint itself. The review would be strengthened by a deeper discussion of this issue: the authors should mention that patient unblinding (due to peppermint’s distinctive smell) and the soothing ritual of inhalation could produce placebo responses. As a limitation, it should be acknowledged that it’s unclear how much of the observed benefit is a direct pharmacologic effect of peppermint oil and how much is a non-specific effect of aroma or breathing. If data allow, a subgroup analysis separating trials with inert placebo controls from those with active controls could be informative – differences in outcomes between these might indicate the degree of placebo effect. At minimum, the authors should caution readers that peppermint’s efficacy might not greatly exceed that of bland aromas or breathing exercises, given the mixed results across control conditions.

Response 3: Thank you for this insightful comment. We agree that the diversity of control interventions across included RCTs introduces important interpretive challenges. In response, we have expanded the Discussion to address these points more explicitly. Relevant additions to the manuscript include:

However, it is important to emphasize that these findings should be considered exploratory, given the substantial methodological limitations and heterogeneity across the included studies. (page 18, lines 468-470)

Additionally, the act of inhalation itself can provide symptom relief, regardless of the aroma, making it unclear how much benefit is due to peppermint oil’s pharmacologic action versus nonspecific effects. (page 20, lines 579-581)

Comments 4: The title of the manuscript – “Inhaling Peppermint Essential Oil is Beneficial in the Treatment of Nausea and Vomiting” – reads as a definitive statement. This could be seen as overstating the findings, considering that the review’s own analysis rates the evidence as low-certainty and notes multiple limitations. It is recommended to moderate the title and similar statements. For example, a more cautious title might be “…Potentially Beneficial…” or “…A Promising Complementary Therapy…”, which the authors actually use in the abstract conclusion . Similarly, in the Conclusion section of the paper and the Abstract, the language should reflect that the results suggest a potential benefit rather than an established one. The authors do note in the Discussion and Conclusions that the evidence is limited (e.g. “the certainty of evidence is low” ), but this caution is somewhat undermined by the unequivocal phrasing in the title and some summary statements. Aligning the tone throughout the manuscript to be consistently careful would enhance credibility. The journal’s readership should not be misled to believe peppermint inhalation is a proven therapy; rather, it should be presented as an interesting adjunct that shows promise but requires further high-quality confirmation.

Response 4: Thank you for your thoughtful feedback regarding the tone and phrasing of the title, abstract, and conclusion. We agree that a more cautious and nuanced presentation is warranted to accurately reflect the limitations and low certainty of the evidence.

We revised the title to: Inhaling Peppermint Essential Oil is a Promising Complementary Therapy in the Treatment of Nausea and Vomiting (page 1 lines 2-3)

Changes in the abstract results and conclusion:

Results: Nineteen RCTs were included. In three PONV studies showed that peppermint oil inhalation was associated with a significantly reduction in nausea 2-6 hours after intervention (MD: –0.60 points, 95% confidence interval (CI): –0.77 to –0.44, p = 0.004). In three NVP studies, showed that daily peppermint oil treatment was linked to lower reduced symptom severity at 48 hours (MD: –0.51, 95% CI: –0.78 to –0.24, p = 0.015) and 96 hours (MD: –0.68, 95% CI: –1.09 to –0.27, p = 0.019). In three CINV studies, showed that peppermint oil inhalation appeared to reduce symptoms at all time points, with the most notable significant reduction at 48 hours (MD: –2.23, 95% CI: –3.13 to –1.34, p < 0.001) and 72 hours (MD: –2.41, 95% CI: –3.96 to –0.86, p = 0.010).

Conclusion: Peppermint oil inhalation appears to may be a promising complementary therapy for reducing nausea and vomiting in postoperative, chemotherapy, and pregnancy settings. (page 1, lines 31-41)

Changes in the conclusion:

This systematic review and meta-analysis indicate suggest that peppermint oil inhalation is may offer a potential complementary approach for managing nausea and vomiting. However, the certainty of evidence is low due to methodological limitations, inconsistency, and imprecision. In postoperative patients, peppermint oil inhalation is appeared most effective in reducing nausea and vomiting severity within 2–6 hours after the intervention, with variable effects at other time points. In pregnant women, peppermint oil inhalation significantly was associated with reduced severity of nausea and vomiting at 48 and 96 hours after intervention, though the evidence in this group is limited by small sample sizes and study heterogeneity. In chemotherapy patients, peppermint oil inhalation consistently reduced the severity of nausea and vomiting at all analyzed time points, with the most significant notable effects observed at 48 and 72 hours, and sustained benefit through 96 hours. Nevertheless, these findings are based on low-certainty evidence. (page 21, lines 627-638)

Comments 5: The review focuses on efficacy (reduction in nausea/vomiting scores) but provides little detail on adverse events or safety of peppermint oil inhalation. This is especially important given the patient populations (e.g. pregnant women, postoperative patients) where safety is paramount. In the Discussion, the authors briefly mention that peppermint oil is a natural therapy with minimal adverse effects and that it is considered safe in pregnancy (category B2) per one reference . However, the manuscript does not systematically report any side effects observed in the trials. Were there any adverse events or complaints (such as allergic reactions, heartburn, headaches, etc.) reported in the RCTs included? If the included studies tracked these, the review should summarize them. Even if no significant adverse outcomes were noted, stating this explicitly is valuable. Conversely, if safety data were not reported by many trials, that is a limitation to acknowledge (potential publication bias towards reporting only benefits). The authors should consider adding a brief results subsection on safety/tolerability. This would reassure readers that peppermint inhalation, as suggested, is indeed low-risk, and it would align with the claim that it’s a “safer option” for patients .

Response 5: Thank you for highlighting the importance of systematically reporting adverse events and safety data. We fully agree that a clear summary of safety and tolerability is essential to support that peppermint oil inhalation is a low-risk, complementary therapy.

We have carefully reviewed all included RCTs for information on adverse events and have summarized the available data. Unfortunately, most included studies did not provide further details regarding safety than what we already included. (page 15, lines 371-381)

Comments 6: In the Introduction, the manuscript states that PONV occurs in about one-third of patients and “even more frequently in patients exposed to high-risk factors such as older age, co-morbidity, obesity, and alcohol consumption, as well as smoking…” . This appears to conflict with established knowledge. Typically, youngerage (not older) and non-smoking status (not smoking) are known patient-specific risk factors for PONV . In fact, smoking and older age have been associated with lower incidence of PONV in some risk scores . It seems the authors may have misinterpreted sources or made a typo in listing risk factors. They cite references [2] and [3] for these claims, but reference 3 is an RCT about peppermint aromatherapy (not a source on PONV risk factors). This should be corrected for accuracy. The authors should revise that sentence to reflect correct risk factors (e.g. younger age, female sex, non-smoker, history of PONV, use of opioids, etc.) or cite a proper reference that supports the risk factors listed.

Response 6: Thank you for your careful review and for highlighting this important point. We agree that the original list of risk factors did not accurately reflect the current evidence and that reference 3 was not appropriate for this statement.

We removed reference 3 and modified the text as: They occur in approximately one out of three postoperative patient and are even more frequent in those with high-risk factors, such as older age, co-morbidity, obesity, and alcohol consumption, as well as smoking, medications female gender, non-smoking status, motion sickness, history of PONV, or the use of opioid analgesics. (page 2, lines 59-62).

Comments 7: A few sentences could be clarified or grammatically adjusted for better readability. For example, the Introduction contains the line: “Although several randomized controlled trials (RCTs) have been conducted to evaluate the effect of peppermint on PONV, NVP, and CINV; its clinical efficacy has not been established yet [20-25].” . The use of a semicolon after “CINV” is not standard because the clause after the semicolon is dependent on the “Although” clause. It would be smoother to split this into two sentences or use a comma: e.g. “Although several RCTs have evaluated peppermint oil for PONV, CINV, and NVP, its clinical efficacy has yet to be established [20–25].”

Similarly, in section 4.2 (Implications for Practice and Research), there is a duplicated sentence about the need for larger studies in NVP: “Larger studies with measurements at multiple time points are needed to better understand the impact of the intervention at different stages of pregnancy.” – this exact sentence appears twice in succession . The authors should remove the redundant sentence. Aside from these, the overall writing is clear.

Response 7: Thank you for your helpful suggestions regarding sentence structure and clarity. We revised the sentence in the Introduction, the text now reads:

Although Several randomized controlled trials (RCTs) have been conducted to evaluate the effect of peppermint on PONV, NVP, and CINV. However, its clinical efficacy has not been established yet [29-34] (page 2, lines 93-95).

Thank you for noticing the duplicate sentence. We removed it (page 21, lines 622-624).

Comments 8: The text should ensure proper spacing and notation around numeric values, units, and confidence intervals. In the Abstract Results (and elsewhere), there are instances of missing spaces, for example: “(MD:-0.60points, 95%confidence interval (CI):-0.77 to -0.44, p=0.004)” . This should be formatted as “MD = –0.60 points, 95% confidence interval (CI): –0.77 to –0.44, p = 0.004.” for clarity. Similar spacing issues occur with other values (e.g., “MD:-2.23, 95%CI:-3.13 to -1.34” should have spaces after the colon). These are minor typographical details but are important for professionalism and readability. The authors should review all statistical reporting to ensure consistency with journal style (e.g. including leading zeroes, using en-dashes for negative signs consistently, etc.).

Response 8: Thank you for your valuable attention to detail regarding the formatting of numeric values, units, and confidence intervals throughout the manuscript. We have carefully reviewed the entire manuscript to identify and correct all instances of missing spaces around numeric values, units, and statistical notations. Also, we have standardized the formatting of statistical results

Comments 9: The review mentions “peppermint spirit” in a couple of instances when discussing prior studies . It may be unclear to readers what “peppermint spirit” refers to – presumably an ethanol-based peppermint preparation used in older studies or clinical practice. Since this term is not widely known, consider adding a brief explanation in the text or a footnote. For example, when first mentioning peppermint spirit, the authors could note it is an alcohol-based solution of peppermint oil traditionally used for nausea relief. This will help avoid confusion, as readers might think “spirit” is a typo or a completely different substance. Consistently using the term “peppermint essential oil (peppermint EO)” elsewhere but then referencing “peppermint spirit” could be clarified for consistency.

Response 9: Thank you for your helpful suggestion regarding the clarity of terminology. We have revised the manuscript to provide a clear explanation of “peppermint spirit” at its first mention, ensuring readers understand the distinction from peppermint essential oil. The revised text:

Most studies used peppermint essential oil as an intervention, with a few using peppermint spirit. Peppermint spirit is a pharmacy-grade, alcohol-based solution containing approximately 82% ethyl alcohol, peppermint oil, peppermint leaf extract, and purified water [58, 59]. (page 6, lines 232-234)

Comments 10: The manuscript references several figures and tables in the Appendix/Supplement (e.g., Figures S9–S14 for risk of bias, Tables S3–S5 for extended characteristics) which is appropriate. One minor point is to ensure that all these supplementary items are correctly labeled and cited in the text. For instance, Table 1 in the main text summarizes basic study characteristics, and the text says extended data is in Table S2 (likely a typo, since later it mentions Tables S3–S5 for subgroups) . Double-check the numbering of supplementary tables/figures and their call-outs to avoid confusion. Also, if possible, ensure that important outcomes like risk-of-bias summaries or GRADE assessments that are relegated to supplements are at least briefly described in the main text (the authors do mention them in words, which is good). No inconsistencies were noted in the results between text and tables as far as can be seen, but the authors should carefully proofread numbers (mean differences, CIs, p-values) to ensure they match between the abstract, main text, and any figures. Any minor discrepancies should be corrected.

Response 10: Thank you for your careful review and helpful suggestions regarding the supplementary materials and consistency of data presentation. Table S2 is a summary table of all the analyzed articles about the types and original ranges of measurement tools evaluating nausea and vomiting. As there were different measurement tools and scales and we uniformized them to be able to analyze, we found important to also include somewhere the original scales. Meanwhile Table S3-S5 are individual tables for the subgroups about further basic characteristics information.

Comments 11: The manuscript references several figures and tables in the Appendix/Supplement (e.g., Figures S9–S14 for risk of bias, Tables S3–S5 for extended characteristics) which is appropriate. One minor point is to ensure that all these supplementary items are correctly labeled and cited in the text. For instance, Table 1 in the main text summarizes basic study characteristics, and the text says extended data is in Table S2 (likely a typo, since later it mentions Tables S3–S5 for subgroups) . Double-check the numbering of supplementary tables/figures and their call-outs to avoid confusion. Also, if possible, ensure that important outcomes like risk-of-bias summaries or GRADE assessments that are relegated to supplements are at least briefly described in the main text (the authors do mention them in words, which is good). No inconsistencies were noted in the results between text and tables as far as can be seen, but the authors should carefully proofread numbers (mean differences, CIs, p-values) to ensure they match between the abstract, main text, and any figures. Any minor discrepancies should be corrected.

Response 11: Thank you for your helpful suggestions, to enhance transparency, we have expanded the main text to briefly describe the key findings from the risk-of-bias and GRADE assessments:

The risk of bias was a concern across all studies. While some studies were well-conducted, most had issues related to blinding participants and personnel be-cause of peppermint oil’s distinctive scent. Only a few postoperative studies achieved a low risk of bias across all domains, and pregnancy studies also had some concerns. In chemotherapy studies, intervention adherence and reporting were generally robust, but uncertainties in randomization remained. A summary of the risk of bias assessment is presented in Figures S9–S14. (page 17, lines 427-433)

The GRADE assessment further shows that the certainty of evidence for peppermint oil inhalation in these settings were consistently low, mainly due to risk of bias and imprecision from small sample sizes. Although different measurement tools con-tributed to some heterogeneity, inconsistency and indirectness were not major concerns. Overall, while peppermint oil inhalation may help reduce nausea and vomiting, confidence in these findings is limited. A summary of the GRADE findings is presented in Table 2. (page 17, lines 434-439)

These limitations highlight the need for cautious interpretation of the pooled results and underscore the importance of rigorous study design, including improved randomization, effective blinding, and comprehensive reporting. Larger, well-designed trials with standardized outcome measures and better blinding are needed in future studies. (page 17- lines 440-443)

Comments 12: Lack of discussion on clinical importance of effects: As noted in the major concerns, the manuscript does not acknowledge that statistically significant improvements might not equate to clinically meaningful benefits. This is a limitation of the evidence synthesis – if an intervention’s effect size is below the threshold of patient-noticeable improvement, its practical value is limited. The authors should explicitly acknowledge this point as a limitation. Currently, the discussion focuses on statistical outcomes and evidence certainty, but not on whether the degree of nausea reduction would matter to patients.

Response 12: Please check Response 2.

Comments 13: Unaddressed placebo and expectation effects: While the authors discuss heterogeneity due to different comparators, they stop short of explicitly noting that the open-label nature of aromatherapy trials and active control designs could mean the observed benefits are partly placebo effects. This is a potential weakness of all aromatherapy studies: patients know they are receiving a fragrant intervention, which could relax them or create positive expectation of relief. The manuscript does not directly call this out as a limitation. It would strengthen the critical appraisal if the authors mention that due to lack of blinding, some proportion of the effect could be non-specific. Acknowledging this would not diminish the findings, but would frame them appropriately (i.e. peppermint oil’s observed efficacy includes any placebo effect inherent to aromatherapy).

Response 13: Thank you for this insightful comment. We agree that the open-label nature of most aromatherapy studies, including those in our review, introduces the potential for placebo and expectation effects, which may contribute to the observed benefits. We have now addressed this limitation in both the Strengths and Limitations section and the Discussion:

Furthermore, the open-label nature of most aromatherapy studies makes it possible that some observed benefits are due to placebo or expectation effects. Since patients might be aware that they are receiving a fragrant intervention, part of the effect may reflect relaxation or positive expectations rather than a direct pharmacologic action of peppermint oil. (pages 20, lines 580-586)

Additionally, as most included studies lacked blinding, patient awareness of receiving a fragrant intervention could enhance placebo responses or create positive expectations of symptom relief. (page 18, lines 484-486)

Comments 14: No aggregation of safety data: The review did not compile or report adverse event data from the trials, and the authors did not list this omission as a limitation. If none of the studies reported safety outcomes, that itself is a limitation to note (because it means we have limited evidence on harms). If studies did report it and the authors chose not to meta-analyze or describe it, that is an oversight. Either way, not addressing safety outcomes in the results is a gap. The authors should acknowledge that “potential side effects were not systematically analyzed” if applicable. This would inform readers that, for example, issues like allergic reactions or other side effects were outside the scope or rarely reported. Given that the context includes pregnant patients, the absence of documented safety data should be highlighted as a limitation in applying these results.

Response 14: Please check Response 5.

Comments 15: Improve sentences that currently read awkwardly. For instance, the use of “;” after an introductory “Although…” (Introduction, lines 75–77) is incorrect – splitting into two sentences or using a comma would fix this. Ensure dependent clauses are properly connected to main clauses. Also, check for any overly long sentences that could be broken up for clarity. Most of the text is concise, but a careful read-through could spot places to simplify phrasing.

Remove the repeated sentence in the Implications for Practice section regarding NVP (as noted above, the line about “Larger studies with multiple time points…” appears twice back-to-back ). Also, verify that no other passages are unintentionally duplicated.

Response 15: Please check response 7.

Comments 16: The manuscript mostly uses past tense for the review’s methods and present tense for discussing results, which is appropriate. Just maintain consistency (e.g., sometimes “was” vs “is” in describing known facts can be mixed). The tone is appropriately formal and scientific. One suggestion is to ensure cautious language where needed (see major concern about not overstating findings). Phrases like “appears to” or “may” are used in several places – continue using those to avoid definitives that the data don’t support strongly.

Response 16: Thank you for your helpful feedback regarding tense consistency and the use of cautious language. We have carefully reviewed the entire manuscript to ensure consistent use of the past tense for methods and the present tense for results and established facts. Additionally, we have ensured that cautious language is consistently used when interpreting findings, in order to avoid overstating the strength of the evidence

Comments 17: Fix small typographical errors such as missing spaces (e.g., in “95%CI” as mentioned), inconsistent use of hyphens/en-dashes, or any misspelled words. We noted the typo “To What Extend” in reference 50 (should be “Extent”) – likely a typo in the reference list (see below). Such minor errors should be corrected to improve overall professionalism.

Response 17: Please check Response 8. We have verified that this is indeed the original title of the cited article.

Comments 18: Reference formatting consistency: The references need some editing to adhere to journal format. For example, Reference #1 (Zhong et al.) is listed as “Int J Mol Sci. 2021, 11, 22.” , which is confusing/mislabeled – International Journal of Molecular Sciences volume 22 (issue 11) in 2021 should likely be formatted as 2021, 22(11), [page or article ID]. It appears the volume/issue may be reversed or an article number is missing. Likewise, Reference #50 (Eghbali et al.) ends with “J Hematol Thromboemb Dis. 2017, 05.” – the “05” is presumably the volume number, but it should not have a leading zero and should include issue/pages or an article ID. The authors should review each reference for completeness: ensure every citation has Journal name, year, volume, and page range or e-location as required by the journal. Several references in the list have all components (e.g., most of the RCT references are complete), but a few like #1 and #50 seem incomplete or oddly formatted. This needs correction in the final bibliography.

Response 18: Thank you for your careful review and for highlighting the need for consistent and complete reference formatting. In response, we have thoroughly reviewed the entire reference list to ensure that each citation adheres to the journal’s required format.

Comments 19: Reference title typos and capitalization: In reference #50, the title is given as “To What Extend Aromatherapy with Peppermint Oil Effects on Chemotherapy Induced Nausea and Vomiting in Patient Diagnosed with Breast Cancer? A Randomized Controlled Trial.” . This looks like a typo (“Extend” should be “Extent”, “Effects” should likely be “Affects” or “Effects on…?”). It’s possible the original article had a grammatical error in the title, but more likely it’s a transcription mistake. The authors should verify the exact title of that source and correct the spelling (“Extent”) and wording if necessary. Additionally, ensure consistent capitalization in titles (most seem fine, just make sure things like acronyms or proper nouns are appropriately capitalized and others are not, according to journal style).

Response 19: Please check comment 17.

Comments 20: Punctuation and spacing in references: Some reference entries are missing a period at the end or have inconsistent punctuation. For instance, reference #49 ends with page “90-104” without a period in the text we see (the format should typically include a period at the end of each citation). Reference #60 lists an article ID (e118983) – which is fine – just ensure the formatting around it is consistent (it appears as 2024, 218, e118983which is correct for a volume and article number, but double-check spacing and punctuation). These are minor formatting quibbles, but since JCM/MDPI has specific reference style, it’s worth carefully proofreading the entire reference list.

Response 21: Thank you for your attention to detail regarding punctuation and spacing in the reference list. We have thoroughly reviewed the entire reference list to ensure consistency with the journal’s required style.

Comments 22: Appropriateness of citations for statements: The majority of in-text citations appear appropriate and up-to-date. The authors have done well to include recent studies (even 2024 publications) and relevant prior reviews . However, there is one notable mismatch: in the Introduction, as mentioned, references [2] and [3] are cited for risk factors of PONV , but reference [3] is not about risk factors. It’s unusual to cite an aromatherapy RCT (Sites et al. 2014) in the middle of a sentence about “older age, co-morbidity, smoking” being risk factors. This likely indicates a citation error (perhaps the authors intended to cite a different source for PONV risk factors). The authors should correct this by citing an authoritative source (e.g., an anesthesiology review or guidelines) for PONV risk factors in that spot, instead of the RCT. Ensuring each statement is backed by a relevant citation is crucial. Another example to double-check: the statement “peppermint oil blocks serotonin receptors in the gut” is backed by reference [17] – the authors should ensure reference 17 indeed supports that mechanistic claim (assuming it’s a pharmacological study on menthol). These are minor content-to-citation alignments to verify.

Response 22: Thank you for your careful review and for highlighting the importance of citation accuracy and alignment. We have corrected the citation error in the Introduction. The Sites et al. (2014) reference has been removed from the sentence listing risk factors for PONV. Instead, we have cited more relevant articles. (page 2, lines 59-62)

We have also verified and updated the reference regarding peppermint oil and serotonin receptors. (page 2, lines 88-89)

Comments 23: Citation order and completeness: The reference numbering appears to be in reasonable order of appearance. We did not find any obviously uncited references in the list. One suggestion is to ensure all included RCTs in the meta-analysis are cited in the text (most are cited in the Results/Discussion when describing individual study findings, which is good). If any study was included but not mentioned, it might be useful to cite it when discussing the results for completeness. The authors might consider adding a PRISMA flow diagram (Figure 1 was likely such, as noted ) which helps in referencing how many studies were included/excluded – presumably they have that. As long as all 19 RCTs are properly referenced somewhere (which, given references like [20–25], [43–48] etc., seems to be the case), that is fine.

Response 23: Thank you for your comment regarding citation order and completeness. We have carefully reviewed the manuscript to ensure that all included RCTs in the meta-analysis are properly cited in the text.

Round 2

Reviewer 2 Report (New Reviewer)

Comments and Suggestions for Authors

The authors had addressed all my comments. I had no further comments to be provided.

Author Response

1. Summary

Thank you very much for taking the time to review this manuscript one more time. Please find the point-by-point responses to your comments below.

2. Point-by-point response to Comments and Suggestions for Authors

Comments 1: The authors had addressed all my comments. I had no further comments to be provided.

Response 1: We sincerely thank you for your thoughtful and constructive feedback throughout the review process. Thank you for your time and valuable input, which have significantly contributed to strengthening our work.

Reviewer 4 Report (New Reviewer)

Comments and Suggestions for Authors

Detailed Assessment by Priority
HIGH PRIORITY UNADDRESSED ISSUES
1. Sensitivity Analyses Not Performed (Comment 1)
Reviewer Request: Perform sensitivity analyses excluding high risk-of-bias studies to test whether findings are driven by biased results.

Author Response: Acknowledged the need but did not perform any sensitivity analyses.

Assessment: This is a fundamental methodological requirement for meta-analyses with heterogeneous, low-quality evidence. The absence of sensitivity analyses significantly undermines confidence in the robustness of the findings.

Recommendation: Authors must either:

    •    Conduct sensitivity analyses excluding high risk-of-bias studies
    •    Provide compelling methodological justification for why sensitivity analyses cannot be performed
    •    Acknowledge this as a major limitation affecting interpretation

2. Systematic Safety Data Analysis Missing (Comments 5 & 14)
Reviewer Request: Systematically compile and report adverse events from included trials, particularly important for vulnerable populations (pregnant women, postoperative patients).

Author Response: Superficial acknowledgment that most studies didn’t provide safety details without attempting systematic extraction.

Assessment: Safety reporting is crucial for clinical recommendations, especially for vulnerable populations. The cursory response fails to meet basic standards for systematic review methodology.

Recommendation: Authors should:

    •    Systematically extract all available safety data from individual studies
    •    Create a dedicated safety/tolerability results subsection
    •    If insufficient data exists, explicitly acknowledge this as a major limitation affecting clinical applicability
    •    Discuss implications for different patient populations

MEDIUM-HIGH PRIORITY UNADDRESSED ISSUES
3. Missing Subgroup Analysis for Control Types (Comment 3)
Reviewer Request: Analyze trials with inert placebo controls separately from those with active controls to assess the degree of placebo effect.

Author Response: General acknowledgment of control variability but no subgroup analysis performed.

Assessment: This analysis is crucial for understanding whether observed benefits are due to peppermint’s pharmacological effects or general therapeutic context. The Sites et al. study showing no difference between peppermint + breathing vs. breathing alone was not adequately addressed.

Recommendation: Perform subgroup analysis comparing:

    •    Studies with inert placebo controls
    •    Studies with active controls (breathing exercises, other aromatherapy)
    •    Discuss implications for interpreting peppermint’s specific efficacy

MEDIUM PRIORITY ISSUES
4. Incomplete Risk Factor Correction (Comment 6)
Reviewer Request: Correct incorrectly stated PONV risk factors (older age and smoking should be younger age and non-smoking).

Author Response: Partial correction but revised text still contains questionable elements and mixed accuracy.

Assessment: The correction appears incomplete and may still contain inaccuracies. This affects the credibility of the background information.

Recommendation: Provide a complete, accurate list of established PONV risk factors with appropriate citations from authoritative sources.

5. Insufficient Evidence for Clinical Context Claims (Comment 2)
Reviewer Request: Provide evidence supporting claims that sub-MCID improvements are clinically meaningful.

Author Response: General statements about complementary therapy value without supporting citations or evidence.

Assessment: While the authors appropriately acknowledged effect sizes below MCID, their claims about clinical relevance lack empirical support.

Recommendation: Either:

    •    Provide citations supporting the clinical value of sub-MCID improvements in these specific contexts
    •    Remove unsupported claims about clinical meaningfulness
    •    Frame as hypothesis requiring future research

6. Persistent Writing and Grammar Issues (Comments 7 & 15)
Reviewer Request: Fix grammatical errors and improve writing clarity.

Author Response: Partial corrections but some issues remain unresolved.

Assessment: Some grammatical problems persist (e.g., “Although Several randomized controlled trials… However, its clinical efficacy…”).

Recommendation: Comprehensive proofreading and editing for grammatical correctness and clarity.

Author Response

Response to Reviewer 4 Comments

1. Summary

Thank you very much for taking the time to review this manuscript one more time. Please find the point-by-point responses to your comments below.

2. Point-by-point response to Comments and Suggestions for Authors

Comments 1: Sensitivity Analyses Not Performed (Comment 1)

Reviewer Request: Perform sensitivity analyses excluding high risk-of-bias studies to test whether findings are driven by biased results.

Author Response: Acknowledged the need but did not perform any sensitivity analyses.

Assessment: This is a fundamental methodological requirement for meta-analyses with heterogeneous, low-quality evidence. The absence of sensitivity analyses significantly undermines confidence in the robustness of the findings.

Recommendation: Authors must either:

    •    Conduct sensitivity analyses excluding high risk-of-bias studies

    •    Provide compelling methodological justification for why sensitivity analyses cannot be performed

    •    Acknowledge this as a major limitation affecting interpretation

Response 1: Thank you for highlighting the importance of sensitivity analyses in assessing the robustness of meta-analytic findings, especially in the context of heterogeneous and low-certainty evidence.

Given that only one study (Cetin et al.) in the postoperative group was rated as high risk of bias, we have already performed a separate meta-analysis for postoperative outcomes, excluding this study. The results of this analysis are presented in the Supplementary Materials (Figure S1) and discussed in the main text:

The study by Cetin et al. 2024 [54] reported a significant difference in the incidence of PONV at baseline, with some concerns about potential bias. Significantly more patients in the control group experienced PONV immediately after surgery than the peppermint oil group (control: 18/38 vs. peppermint: 2/38, p = 0.001). This difference may confound the results. To address this concern, we conducted a sensitivity analysis by excluding the study by Cetin from the meta-analysis. (page 8, lines 293–298)

Comments 2: Systematic Safety Data Analysis Missing (Comments 5 & 14)

Reviewer Request: Systematically compile and report adverse events from included trials, particularly important for vulnerable populations (pregnant women, postoperative patients).

Author Response: Superficial acknowledgment that most studies didn’t provide safety details without attempting systematic extraction.

Assessment: Safety reporting is crucial for clinical recommendations, especially for vulnerable populations. The cursory response fails to meet basic standards for systematic review methodology.

Recommendation: Authors should:

    •    Systematically extract all available safety data from individual studies

    •    Create a dedicated safety/tolerability results subsection

    •    If insufficient data exists, explicitly acknowledge this as a major limitation affecting clinical applicability

    •    Discuss implications for different patient populations

Response 2: Thank you for highlighting the need for a thorough and systematic analysis of safety data, particularly given the clinical importance for vulnerable patient groups.

We already had a specific subsection in the Results that summarized the safety data across all studies:

3.2.5. Adverse events

Table S6 summarizes the reported adverse events. No adverse events were reported in postoperative studies with the inhalation of peppermint oil [28, 33, 51-59]. In chemotherapy studies, few patients reported adverse events. In a study by Ertürk et al. 2021 [30], two of 36 patients in the intervention group had headaches, and 3 experienced increased frequency and severity of nausea. In a study by Mapp et al. 2020 [62], only 1 of the 36 patients reported feeling worse after inhaling peppermint. Chemo-therapy studies by Jafarimanesh et al. 2020 [31] and Eghbali et al. 2017 [60] concluded that a standard dose of peppermint oil does not cause adverse events. The study by Lestari et al. 2017 [61] did not report any adverse events after peppermint treatment. No adverse events were reported in the studies on pregnancy during the inhalation of peppermint oil [29, 32, 63]. (page 15, lines 398–408)

Additionally, we have now systematically extracted all available safety and adverse event data from each included study. This information is now clearly summarized in a dedicated table in the Supplementary (Table S6):

Table S6 Summary of the adverse events

Study

Nausea and vomiting type

Adverse event

Ahmadi et al., 2020 [28]

postoperative

not reported

Maghami et al., 2020 [33]

postoperative

not reported

Ferruggiari et al., 2012 [55]

postoperative

not reported

Aydin et al., 2018 [52]

postoperative

not reported

Lane et al., 2012 [57]

postoperative

not reported

Anderson et al., 2004 [51]

postoperative

not reported

Baek et al., 2024 [53]

postoperative

not reported

Imani et al., 2024 [57]

postoperative

not reported

Cetin et al., 2024 [55]

postoperative

not reported

Sites et al., 2014 [59]

postoperative

not reported

Tate et al., 1997 [60]

postoperative

not reported

Amzajerdi et al., 2022 [30]

pregnancy

not reported

Joulaeerad et al., 2017 [33]

pregnancy

not reported

Pasha et al., 2012 [64]

pregnancy

not reported

Ertürk et al., 2021 [31]

chemotherapy

for two patients peppermint oil caused headache and for three patients the frequency and the severity of nausea increased

Jafarimanesh et al., 2020 [32]

chemotherapy

reported no adverse events

Eghbali et al., 2017 [61]

chemotherapy

reported no adverse events

Mapp et al., 2020 [63]

chemotherapy

one patient reported feeling worse after inhaling peppermint

Lestari et al., 2017 [62]

chemotherapy

not reported

We acknowledge that a major limitation of most of the studies is that they did not systematically report adverse events or tolerability outcomes.

A further significant limitation is the lack of systematic safety reporting in the included studies. Most trials did not provide detailed information on adverse events or tolerability. Future studies should systematically collect and report safety data. (page 21, lines 634-637)

Comments 3: Missing Subgroup Analysis for Control Types (Comment 3)

Reviewer Request: Analyze trials with inert placebo controls separately from those with active controls to assess the degree of placebo effect.

Author Response: General acknowledgment of control variability but no subgroup analysis performed.

Assessment: This analysis is crucial for understanding whether observed benefits are due to peppermint’s pharmacological effects or general therapeutic context. The Sites et al. study showing no difference between peppermint + breathing vs. breathing alone was not adequately addressed.

Recommendation: Perform subgroup analysis comparing:

    •    Studies with inert placebo controls

    •    Studies with active controls (breathing exercises, other aromatherapy)

    •    Discuss implications for interpreting peppermint’s specific efficacy

Response 3: We appreciate the reviewer’s suggestion to conduct a subgroup analysis comparing trials with inert placebo controls to those with active controls. Methodologically, such an analysis would indeed be valuable for distinguishing between pharmacological and non-specific therapeutic effects.

After a thorough analytical review of all included studies, we confirm that none of the postoperative and pregancy-related studies used active control. Chemotherapy related studies employed routine antiemetic treatment in both intervention and control groups coupled with the inhalation treatment or placebo, respectively. that none of the trials used active controls. Two articles—Cetin and Ertürk—did not use a control group. All other studies with a comparator group employed inert placebo controls (e.g., saline, distilled water, or no-odour substances like almond oil) that, based on current evidence and our assessment, have no known effect on nausea and vomiting.

For the Cetin article, we conducted a separate meta-analysis (see Figure S1 in Supplementary Material). Comparing analyses including and excluding Cetin et al. showed no statistically or clinically significant difference in the results. In the case of Ertürk et al., we acknowledged the issue that no intervention was used in the control group, whereas all other included studies had a placebo treatment as control:

The study by Ertürk et al. 2021 [30] showed significantly lower nausea scores in the peppermint oil group using different chemotherapy regimens. It is important to note that in this study, the control group did not receive anything, which limits the ability to directly attribute the observed effects to peppermint oil compared to standard care or placebo. However, the studies of Jafarimanesh et al. 2020 [31] and the Eghbali et al. 2017 [60] studies used a placebo control, and they showed smaller differences between the two groups. (page 13 lines, 365-371)

We acknowledged also the importance of the Sites et al. (2014) study, which compared controlled breathing alone to controlled breathing with peppermint spirit for postoperative nausea and vomiting. This study found no significant difference in effectiveness between the two arms, suggesting that much of the observed benefit may be attributable to non-specific effects such as the act of inhalation or relaxation, rather than a direct pharmacologic effect of peppermint oil. We have discussed this point in the manuscript:

Some trials used inert placebos, while others used no controls. The Sites et al. (2014) study [58], for example, found no significant difference in effectiveness between controlled breathing alone and controlled breathing with peppermint spirit for postoperative nausea and vomiting, suggesting that much of the observed benefit may be attributable to non-specific effects such as the act of inhalation or relaxation, rather than a direct pharmacologic effect of peppermint oil. The limited number of studies in each control category prevented meaningful subgroup analysis, further restricting our ability to distinguish between specific and non-specific effects. (page 20, lines 615-623).

Given the small number of studies and limited sample sizes in each control category, we did not wish to pursue further subgroup analysis by control type, as this would result in underpowered and potentially misleading results. We have emphasized in the manuscript that these findings should be interpreted with caution, and we agree that further well-designed clinical trials with appropriate control groups are needed to clarify the specific efficacy of peppermint oil inhalation.

The limited number of studies in each control category prevented meaningful subgroup analysis, further restricting our ability to distinguish between specific and non-specific effects. (page 20, lines 621-623).

The pooled effect estimates could be overestimated due to systematic biases and the limitations described above. We interpret our findings with caution and recommend that future research prioritize rigorous study design, adequate blinding, and standardized outcome measures to improve the reliability of evidence in this field. (page 21, lines 644-648)

Comments 4: Incomplete Risk Factor Correction (Comment 6)

Reviewer Request: Correct incorrectly stated PONV risk factors (older age and smoking should be younger age and non-smoking).

Author Response: Partial correction but revised text still contains questionable elements and mixed accuracy.

Assessment: The correction appears incomplete and may still contain inaccuracies. This affects the credibility of the background information.

Recommendation: Provide a complete, accurate list of established PONV risk factors with appropriate citations from authoritative sources.

Response 4: Thank you for your helpful feedback regarding the accuracy of the risk factors for postoperative nausea and vomiting (PONV) in our introduction. In accordance with your recommendation and based on authoritative sources, we have revised this section to provide a complete and accurate list of established PONV risk factors.

Postoperative nausea and vomiting (PONV) are common adverse events of anesthesia and surgery, occurring in approximately one out of three postoperative patients. The risk is higher in those with established risk factors, including female sex, non-smoking status, a history of PONV, motion sickness, or the use of postoperative opioids. Younger age is also associated with increased risk, with risk decreasing as age advances. [2-4]. The type of surgery and the type of anesthesia, particularly the use of volatile anesthetics, further increase risk, while regional techniques are generally associated with a lower incidence of PONV [5-7]. In particular, for patients undergoing vitreoretinal surgeries, especially those with diabetes, PONV can lead to serious complications such as suprachoroidal hemorrhage, potentially resulting in vision impairment [8, 9]. The relationship between obesity and PONV is controversial. Most evidence indicates that obese patients are not at increased risk and may experience a lower incidence, while underweight patients (BMI < 19) may have a higher risk. [10, 11]. The incidence of PONV may be further reduced by employing Surgical Pleth Index-guided opioid-free anaesthesia [12], or even lowered below 10% with Adequacy of Anaesthesia guidance [13]. (page 2 lines 61-76)

Comments 5: Insufficient Evidence for Clinical Context Claims (Comment 2)

Reviewer Request: Provide evidence supporting claims that sub-MCID improvements are clinically meaningful.

Author Response: General statements about complementary therapy value without supporting citations or evidence.

Assessment: While the authors appropriately acknowledged effect sizes below MCID, their claims about clinical relevance lack empirical support.

Recommendation: Either:

    •    Provide citations supporting the clinical value of sub-MCID improvements in these specific contexts

    •    Remove unsupported claims about clinical meaningfulness

    •    Frame as hypothesis requiring future research

Response 5: Thank you for highlighting the need to support or clarify claims regarding the clinical value of symptom improvements that fall below the minimal clinically important difference (MCID) for nausea and vomiting (NV) outcomes. Based on a review of the literature, the following points address the reviewer’s concerns:

While peppermint oil inhalation was associated with statistically significant reductions in PONV severity, the improvement was modest and generally below the established minimal clinically important difference (MCID) of approximately 1.5 points on a 0–10 nausea scale [64]. The clinical relevance of such modest improvements is uncertain. Where conventional antiemetics are unsuitable, even small symptom relief may be valued by some patients, especially given peppermint oil's favorable safety profile, low cost, and ease of administration. This hypothesis warrants further research to determine its practical significance in clinical care. (page 18, lines 518-532)

While some analyses demonstrated statistically significant reductions in symptom severity, these improvements did not reach thresholds considered clinically meaningful for PONV and NVP patients. The effects may be more promising for CINV patients, but further research is needed to clarify the clinical value of these findings. Given its favorable safety profile and accessibility, peppermint oil inhalation could be considered as an adjunct when conventional antiemetic options are limited or unsuitable. (page 22, lines 695-701)

There is currently no direct empirical evidence supporting the clinical value of sub-MCID improvements in PONV, NVP, or CINV. Therefore, any claims about the clinical meaningfulness of such changes should be considered as hypotheses and areas for future research. (page 21, lines 631-634)

Comments 6: Persistent Writing and Grammar Issues (Comments 7 & 15)

Reviewer Request: Fix grammatical errors and improve writing clarity.

Author Response: Partial corrections but some issues remain unresolved.

Assessment: Some grammatical problems persist (e.g., “Although Several randomized controlled trials… However, its clinical efficacy…”).

Recommendation: Comprehensive proofreading and editing for grammatical correctness and clarity.

Response 6: We thank the reviewer for highlighting the need for comprehensive proofreading and improved clarity. We have carefully reviewed the entire manuscript and corrected all grammatical errors and issues with writing clarity. We believe that all outstanding grammatical and writing issues have now been fully addressed.

This manuscript is a resubmission of an earlier submission. The following is a list of the peer review reports and author responses from that submission.

Round 1

Reviewer 1 Report

Comments and Suggestions for Authors

Thank you for your invitation to review this manuscript "Inhaling Peppermint Essential Oil is Beneficial in the Treatment of Nausea and Vomiting". Overall this is a well-designed systematic review as it included all RCTs. This meta-analysis concluded that peppermint oil inhalation appears to be a promising complementary therapy for reducing nausea and vomiting in postoperative, chemotherapy, and pregnancy settings. However, several aspects of the study need improvement to strengthen its scientific impact. First, it would be better to expand the limitations of this study. Since the intrinsic nature of meta-analysis is a retrospective study, we need to highlight this. Second, since the majority of included studies were conducted in specific regions ex. Iran and Turkey, there would be concerns about cultural and healthcare system variability. It would be beneficial to include a section addressing geographic and demographic biases. At last, we have noticed that some studies used placebo and others used standard care. This heterogeneity might affect pooled comparisons and should be addressed.

Author Response

Thank you very much for the time and effort that you have dedicated to providing valuable feedback for this manuscript titled "Inhaling Peppermint Essential Oil is Beneficial in the Treatment of Nausea and Vomiting" to the Journal of Clinical Medicine. Please find the detailed responses below and the corresponding revisions/corrections, both highlighted and in track changes, in the resubmitted document. We have carefully considered all comments and incorporated changes to address the suggestions.

Below is a point-by-point response to the reviewers’ comments.  

Point-by-point response to Comments and Suggestions for Authors

Comments 1: First, it would be better to expand the limitations of this study. Since the intrinsic nature of meta-analysis is a retrospective study, we need to highlight this.

Response 1: Thank you for highlighting the need to discuss the retrospective nature of meta-analyses. We agree that this is an important limitation. We have expanded the limitations section to explicitly state that, as a meta-analysis, our study is retrospective and thus subject to the inherent limitations of such designs, including the reliance on previously published data, potential publication bias, and the inability to control for all confounders present in the original studies. This addition can be found in the Discussion section, page 17, paragraph 4.1, lines 488-491, where we now state: “The retrospective nature of meta-analyses, including the present study, introduces certain limitations. We are dependent on the quality and reporting of the included RCTs, and unmeasured confounders may remain present. Moreover, publication bias and selective reporting cannot be entirely excluded.”

Comments 2: Second, since the majority of included studies were conducted in specific regions ex. Iran and Turkey, there would be concerns about cultural and healthcare system variability. It would be beneficial to include a section addressing geographic and demographic biases.

Response 2: We appreciate this important observation. We have expanded the limitations section addressing the potential impact of geographic and demographic biases, noting that most included studies were conducted in Iran, Turkey and the USA. We discuss that cultural practices, healthcare system differences, and population characteristics may limit the generalizability of our findings to other regions. This new content is located in the Discussion section, page 17, paragraph 4.1, lines 483-487, where we now state: “The majority of included studies were conducted in Iran, Turkey, and the USA, which may introduce geographic and demographic biases. Differences in cultural practices, healthcare infrastructure, and patient populations could affect the applicability of our results to other settings. Future research should include more diverse populations to enhance generalizability.”

Comments 3: At last, we have noticed that some studies used placebo and others used standard care. This heterogeneity might affect pooled comparisons and should be addressed.

Response 3: Thank you for this valuable comment. We agree that the use of different control interventions (placebo vs. standard care) introduces heterogeneity. We have now addressed this in the Discussion sections, explicitly acknowledging this as a source of heterogeneity and discussing its potential impact on pooled effect estimates. The relevant text can be found in the Discussion, page 17, paragraph 4.1, lines 477-482, where we now state: “Furthermore, studies used different types of control interventions, including placebo (normal saline, distilled water), standard care (routine antiemetics, routine nursing care), and other comparators (alternative oils, controlled breathing). This heterogeneity may lead to potential confounding, as patients receiving standard care may have had access to additional supportive therapies not available in placebo-controlled trials.”

Reviewer 2 Report

Comments and Suggestions for Authors

This paper is a meta-analysis that examines whether peppermint oil is effective against nausea, but I believe that the mechanisms of nausea and patient backgrounds differ for PONV, CINV, and NPV.

Therefore, I concerned that the reliability of the study is not high when comparing patients with different background factors.

In addition, a meta-analysis was conducted using NRS and VAS, and  NRS is evaluated on a value scale of 0-10 and VAS is evaluated on a value scale of 0-100.
Is it acceptable to conduct this analysis using outcomes that have different numbers?

Author Response

Thank you very much for the time and effort that you have dedicated to providing valuable feedback for this manuscript titled "Inhaling Peppermint Essential Oil is Beneficial in the Treatment of Nausea and Vomiting" to the Journal of Clinical Medicine. Please find the detailed responses below and the corresponding revisions/corrections, both highlighted and in track changes, in the resubmitted document. We have carefully considered all comments and incorporated changes to address the suggestions.

Below is a point-by-point response to the reviewers’ comments.  

Point-by-point response to Comments and Suggestions for Authors

Comments 1: This paper is a meta-analysis that examines whether peppermint oil is effective against nausea, but I believe that the mechanisms of nausea and patient backgrounds differ for PONV, CINV, and NPV. Therefore, I concerned that the reliability of the study is not high when comparing patients with different background factors.

Response 1: Thank you for this important comment. We fully agree that PONV, CINV, and NVP differ in mechanisms and patient backgrounds, and that pooling these groups could introduce significant heterogeneity. To address this, we conducted separate meta-analyses for each patient group, and did not pool results across these distinct populations. The results for each group are presented and interpreted independently throughout the manuscript (PONV: paragraph 3.2.1 and 3.2.4, NVP: paragraph 3.2.2, CINV: 3.2.3). We have revised the Abstract (page 1, lines 30-31), Methods (page 4, paragraph 2.9, lines 149-150) and Discussion (page 15, lines 412-414) to emphasize that our findings are specific to each patient group. We hope this addresses your concern regarding the reliability and validity of our study’s conclusions.

In the Abstract now we state: “Separate meta-analyses were conducted for each patient group using R, focusing on nausea and vomiting severity.”

In the Methods at the Synthesis Methods and Statistical Analysis section now we state: “To account for differences in pathophysiology and patient populations, we conducted separate meta-analyses for PONV, CINV, and NVP.”

In the Discussion now we state: “This meta-analysis is the first to comprehensively evaluate the efficacy of pep-permint oil alone compared to control in three distinct patient populations - PONV, CINV, and NVP - with each group analyzed independently at multiple time points.”

Comments 2: In addition, a meta-analysis was conducted using NRS and VAS, and NRS is evaluated on a value scale of 0-10 and VAS is evaluated on a value scale of 0-100.

Is it acceptable to conduct this analysis using outcomes that have different numbers?

Response 2: Thank you for raising this important methodological question. We agree that combining outcomes measured on different scales (NRS 0–10 vs. VAS 0–100) requires careful handling to ensure validity. To address this, we converted all scales to a standardized 0–10 metric, as described in the Methods section (page 4, paragraph 2.9, lines 149-161). For example, VAS scores (0–100) were divided by 10 to align with the NRS scale (0–10). This approach is widely accepted and consistent with established meta-analytic methods for harmonizing disparate scales and allows for meaningful pooling of mean differences (MDs).

Round 2

Reviewer 2 Report

Comments and Suggestions for Authors

No further comments for this manuscripts.